

# Comparison of mean age of air in five reanalyses using the BASCOE transport model

Simon Chabrillat[1], Corinne Vigouroux[1], Yves Christophe[1], Andreas Engel[2], Quentin Errera[1], Daniele Minganti[1], Beatriz M. Monge-Sanz[3], Arjo Segers[4], and Emmanuel Mahieu[5]

[1]Royal Belgian Institute for Space Aeronomy, BIRA-IASB, Brussels, 1180 Belgium
[2]Institute for Atmospheric and Environmental Science, Goethe University Frankfurt, Frankfurt, Germany
[3]European Centre for Medium-Range Weather Forecasts, Shinfield Park, Reading, RG29AX, UK
[4]TNO, Department of Climate, Air and Sustainability, P.O. box 80015, 3508 TA Utrecht, the Netherlands
[5]Institute of Astrophysics and Geophysics, University of Liège, 4000 Liège, Belgium

*Correspondence to:* Simon Chabrillat
(Simon.Chabrillat@aeronomie.be)

**Abstract.**

We present a consistent intercomparison of the mean Age of Air (AoA) according to five modern reanalyses: the European Centre for Medium-Range Weather Forecasts Interim Reanalysis (ERA-Interim), the Japanese Meteorological Agency's Japanese 55-year Reanalysis (JRA-55), the National Centers for Environmental Prediction Climate Forecast System Reanalysis

(CFSR) and the National Aeronautics and Space Administration's Modern Era Retrospective-analysis for Research Applications version 1 (MERRA) and version 2 (MERRA-2). The modeling tool is a kinematic transport model driven only by the surface pressure and wind fields. It is validated for ERA-I through a comparison with the AoA computed by another transport model.

The five reanalyses deliver AoA which differ in the worst case by one year in the tropical lower stratosphere and more than

10 two years in the upper stratosphere. At all latitudes and altitudes, MERRA-2 and MERRA provide the oldest values (~5-6 years in mid-stratosphere at mid-latitudes) while JRA-55 and CFSR provide the youngest values (~4 years) and ERA-I delivers intermediate results. The spread of AoA at 50 hPa is as large as the spread obtained in a comparison of Chemistry-Climate Models. The differences between tropical and mid-latitudes AoA are in better agreement except for MERRA-2. Compared with in-situ observations, they indicate that the upwelling is too fast in the tropical lower stratosphere. The general hierarchy

of reanalyses delivering older AoA (MERRA, MERRA-2) and younger AoA (JRA-55, CFSR) holds during the whole 1989–2015 period, with AoA derived from ERA-I keeping intermediate values. The spread between the five simulations in the northern mid-latitudes is as large as the observational uncertainties in a multidecadal time series of balloon observations, i.e., approximately two years. No global impact of the Pinatubo eruption can be found in our simulations of AoA, contrarily to a recent study which used a diabatic transport model driven by ERA-I and JRA-55 winds and heating rates.

The time variations are also analyzed through multiple linear regression analyses taking into account the seasonal cycles, the Quasi-Biennal Oscillation and the linear trends over four time periods. The amplitudes of AoA seasonal variations in the lower stratosphere are significantly larger using MERRA and MERRA-2 than with the other reanalyses (up to twice as large at





the 50 hPa pressure level). The linear trends of AoA using ERA-I confirm those found by earlier model studies, especially for the period 2002–2012 where the dipole structure of the latitude-height distribution (positive in the northern mid-stratosphere and negative in the southern mid-stratosphere) also matches trends derived from satellite observations of $SF_6$. Yet the linear trends vary considerably depending on the considered period. Over 2002–2015 the ERA-I results still show a dipole structure but it is much less pronounced, with positive trends in the northern hemisphere remaining significant only in the polar lower stratosphere (where they reach 0.2 years per decade). No reanalysis other than ERA-I finds any dipole structure of AoA trends. The signs of the trends depend strongly on the input reanalysis and on the considered period, with values above 10 hPa varying between approximately -0.4 and 0.4 years per decade. Using ERA-I and CFSR, the 2002–2015 trends are negative above 10 hPa but using the three other reanalyses these trends are positive. Over the whole period 1989–2015 each reanalysis delivers opposite trends, i.e., AoA is mostly increasing with CFSR and ERA-I but mostly decreasing with MERRA, JRA-55 and MERRA-2.

In view of these large disagreements, we urge great caution for studies aiming to assess AoA trends derived only from reanalysis winds. We briefly discuss some possible causes for the dependency of AoA on the input reanalysis and highlight the need for complementary intercomparisons using diabatic transport models.

# 1   Introduction

The mean age of air (hereafter AoA) is an evaluation of the time necessary for variations of long-lived (e.g., greenhouse or ozone-depleting) species to propagate from the troposphere to various regions in the stratosphere. This classical diagnostic provides insights on the strength and structure of the Brewer-Dobson Circulation (BDC), the polar vortex, and irreversible mixing in the mid-latitudes (Waugh and Hall, 2002). Due to increased greenhouse gas forcing, Chemistry-Climate Model (CCM) simulations of the 1990-2090 period predict an acceleration of the BDC and a decrease of AoA at all latitudes in the lower part of the stratosphere (Austin and Li, 2006; Butchart, 2014). The observational detection of trends in the BDC strength turns out to be quite difficult. They can be indirectly derived from multidecadal records of stratospheric temperatures but these derivations are indirect and do not yet allow a clear confirmation of the acceleration predicted by CCM, mainly due to an insufficient quality of the temperature observations (Fu et al., 2015; Ossó et al., 2015).

Observation-based AoA is derived from concentration measurements of very long-lived tracers which increase (nearly) monotonously at the surface, such as $CO_2$ or $SF_6$. Multidecadal datasets were compiled from balloon soundings or aircraft flights (e.g., Andrews et al., 2001; Ray et al., 2014, and references therein). The corresponding time series are precise but sparse in time and space, leading to large sampling uncertainties. Global coverage time series have been derived from satellite observations, but the precision is lower. $SF_6$ retrievals from the Michelson Interferometer for Passive Atmospheric Sounding (MIPAS) satellite instrument delivered a continuously updated dataset with global coverage for the period 2002-2012, leading to breakthrough studies about observed AoA and its time variations during this comparatively short period (Stiller et al., 2008; Haenel et al., 2015). The magnitude, distribution and detectability of the AoA trends observed over the past years and decades are currently a topic of intense research (e.g., Engel et al., 2009; Stiller et al., 2012; Mahieu et al., 2014; Engel et al., 2017).





Reanalysis systems combine a global weather forecast model, observations, and an assimilation scheme to provide the best estimates (analyses) of past atmospheric states including surface pressure, temperature, and wind over a long (usually multi-decadal) period. While they are derived from assimilation systems used operationally to deliver weather forecasts, they aim to achieve more consistent variations over long timescales, e.g., avoiding spurious discontinuities and trends. Hence the same

model version and assimilation scheme are used for the whole period and special care is given to the time-varying biases between the assimilated observations (see, e.g., Simmons et al., 2014). The resulting reanalysis datasets provide a multivariate, spatially complete, and coherent record of the global atmospheric circulation.

The Stratosphere–troposphere Processes And their Role in Climate (SPARC) Reanalysis Intercomparison Project (S-RIP) is a coordinated intercomparison of all major global atmospheric reanalyses. Its introductory paper (Fujiwara et al., 2017)

provides an overview of the past and current reanalysis systems and datasets. The present study deals with the five modern reanalyses of surface and satellite data retained in S-RIP: the European Centre for Medium-Range Weather Forecasts Interim Reanalysis (ERA-Interim), the Japanese Meteorological Agency's Japanese 55-year Reanalysis (JRA-55), the National Centers for Environmental Prediction Climate Forecast System Reanalysis (NCEP-CFSR) and the National Aeronautics and Space Administration's Modern Era Retrospective-analysis for Research Applications version 1 (MERRA) and version 2 (MERRA-

2).

The absolute value of AoA and its evolution over the past decades can be derived from the surface pressure and wind fields available in such reanalyses, using either an offline transport model (see, e.g., Chipperfield, 2006) or a chemistry-climate model nudged to the input reanalysis (Kunz et al., 2011; Kovács et al., 2017) to model the transport of inert tracers propagating from the troposphere to the stratosphere. This approach helped to identify shortcomings in the Brewer-Dobson circulation

described by early reanalyses (Meijer et al., 2004; Pawson et al., 2007) and to assess the improvements in the next generation of reanalyses, e.g., from ERA-40 to ERA-Interim (Monge-Sanz et al., 2007; Dee et al., 2011; Monge-Sanz et al., 2012).

Few AoA comparisons have been performed between reanalyses originating from different reanalysis centers. This is mainly due to technical difficulties which are not limited to file formatting issues. While all modern systems use hybrid $\sigma - p$ vertical coordinates (Simmons and Burridge, 1981), each reanalysis comes with a wind field computed on a different grid with different

horizontal and vertical resolutions. Some reanalysis forecast models use spectral dynamical cores (Krishnamurti et al., 2006) while others use finite-volume dynamics (Lin, 2004, see next section for details). A common offline transport model may have difficulties dealing with such different grids because it is usually tailored for a specific family of reanalyses, e.g., using an advection algorithm similar to the dynamical core of the driving reanalysis system or climate model (Strahan and Polansky, 2006).

Section 2 describes the input reanalyses and our modeling tools to explains how these difficulties were circumvented. It also validates our approach with a classical set of observations and with the results of two other transport models, each tailored for its own reanalysis dataset.

The main purpose of this paper is to provide a comparison of the AoA obtained from five modern reanalyses included in the S-RIP project in order to assess their level of agreement or to identify outliers. Its focus is not on detailed comparisons





with observations (which are deferred to a follow-on study) but rather on a consistent intercomparison between the reanalyses through the use of a common transport model.

Section 3 compares the distribution of the AoA obtained from each reanalysis for a reference period and its time evolution in the middle latitudes. Section 4 uses a multiple linear regression model to characterize the time variations of AoA, including

5 an intercomparison of their linear trends for several periods. Section 5 proposes a brief overview of the possible causes for the disagreements between the reanalyses and states the further work required to elucidate these disagreements. Section 6 concludes the paper with a summary of our findings and their implications.

## 2 Methodology

### 2.1 Description and set-up of the offline transport model

Depending on their vertical coordinate system and the reanalysis data used as input, one may distinguish between kinematic and diabatic transport models (Mahowald et al., 2002; Chipperfield, 2006). Diabatic models use isentropic ($\theta$) or hybrid $\sigma - \theta$ vertical coordinates and calculate the vertical transport from diabatic heating rates which may be read from the input reanalysis or re-computed using a separate radiation scheme. Kinematic transport models on the other hand need on input only the surface pressure and horizontal wind fields, relying on mass continuity to derive the vertical mass fluxes. The present study uses the

kinematic transport model developed for the Belgian Assimilation System for Chemical ObsErvations (BASCOE: see Errera et al., 2008; Skachko et al., 2014; Lefever et al., 2015). Its advection module is the Flux-Form Semi-Lagrangian (FFSL) scheme (Lin and Rood, 1996) configured to follow closely the recommendations of Rotman et al. (2001). We briefly summarize here this configuration because it has an important impact on the simulated distribution of AoA in the stratosphere.

The FFSL advection scheme is run on a evenly-spaced latitude-longitude grid with $2° \times 2.5°$ increments. This grid spacing is

20 typical for current simulations of stratospheric chemistry and transport over several decades (Morgenstern et al., 2017). Using the FFSL algorithm, Strahan and Polansky (2006) showed that this is the minimum resolution allowing a realistic representation of the tropical and high latitude mixing barriers. The FFSL algorithm does not require satisfaction of the Courant-Friedrichs-Lewy (CFL) condition in the longitudinal direction, which is a big computational advantage for regular longitude-latitude grids. The time step is set to 30 minutes by default and automatically split into integer fractions in order to satisfy the CFL

condition in the meridional direction. The algorithmic structure of the FFSL scheme allows multiple choices for monotonicity constraints that have implications on the subgrid tracer distribution used to calculate fluxes across cell edges. These choices are made separately in the longitudinal, meridional and vertical directions. Rotman et al. (2001) showed that AoA calculations are very sensitive to the choice of constraint in the vertical direction: realistic results require a positive-definite Piece-wise Parabolic Method, where the constraint on the subgrid distribution is only strong enough to prevent generation of negative

values but overshoots and undershoots are allowed.

In all our simulations the AoA is computed from the distribution of an idealized tracer which increases linearly at the surface. In order to allow quick propagation of this boundary condition to the free troposphere, eddy vertical diffusion is modeled in the lower half of the troposphere with a vertical diffusion coefficient $K_{zz}$ decreasing from an arbitrary value of 10 m$^2$ s$^{-1}$ at





the surface to zero at the pressure level halfway between the surface and the tropopause. There is no other representation of convection in the model nor any explicit mechanism for horizontal diffusion.

## 2.2 Description of the input reanalyses

We compute and compare the AoA in five recent reanalyses which are described in detail by Fujiwara et al. (2017): ERA-Interim (Dee et al., 2011), JRA-55 (Kobayashi et al., 2015), MERRA (Rienecker et al., 2011), MERRA-2 (Gelaro et al., 2017) and NCEP-CFSR (Saha et al., 2010). These data-sets were used over the period January 1980 to December 2015, except for NCEP-CFSR which originally ended in December 2010 and is extended here with the CFSv2 data-set (Climate Forecast System version 2 Saha et al., 2014) from January 2011 to December 2014. Hereafter we use "ERA-I" to refer to ERA-Interim and "CFSR" to refer to the combined NCEP-CFSR reanalyses.

Each reanalysis is available on two vertical grids: the native grid of the underlying atmospheric model (product on "model levels") and an output grid of constant pressures (product interpolated to "pressure levels"). Our simulations are run on the native model levels in order to account for the different vertical resolution of each reanalysis system and also to avoid any interference from the interpolation methods used to deliver the products on constant pressure levels. All reanalysis systems use the hybrid sigma-pressure vertical coordinate with levels extending from the surface up to ~0.266 hPa (~57 km height) in CFSR, 0.1 hPa (~64 km) in ERA-I and JRA-55, or 0.01 hPa (~78 km) in MERRA and MERRA-2. The reader is referred to Fujiwara et al. (2017) for a comparison of the vertical resolutions of the reanalysis systems.

The forecast models use two different frameworks to discretize their primitive variables on the horizontal plane: MERRA and MERRA-2 solve for mass fluxes on a regular latitude-longitude grid (Lin, 2004) while ERA-I, JRA-55 and CFSR use spectral dynamical cores, i.e., they solve for vorticity and divergence expressed on a spherical harmonics basis (e.g., Krishnamurti et al., 2006). Users of the reanalyses often download velocity fields which are derived from the primitive variables and evaluated on varying regular grids: these may be reduced Gaussian grids (ERA-I and JRA-55), regular Gaussian grids (CFSR) or regular latitude-longitude grids (MERRA and MERRA-2). This pre-processing is described in detail in the next subsection.

We use in all cases the analyses valid at 00 h, 06 h, 12 h and 18 h, i.e., datasets with a 6-h time resolution. The assimilation procedure for MERRA and MERRA-2 uses an iterative predictor–corrector approach, generating two separate sets of reanalysis products designated "ANA" for analysis state and "ASM" for assimilated state (Rienecker et al., 2011). The latter products use a 6h "corrector" forecast centered on the analysis time and an incremental analysis update to apply the previously calculated assimilation increment gradually rather than abruptly at the analysis time (Bloom et al., 1996). Thanks to this procedure, the ASM products have smaller wind imbalances than the ANA products (Fujiwara et al., 2017) hence they are preferable for tracer transport simulations. We used the ASM products in MERRA-2 but could not do so with MERRA where the ASM products are only available on constant pressure levels. Since we aim to evaluate each reanalysis on its native vertical grid, we had to fall back on the ANA product in the case of MERRA.



### 2.3 Pre-processing of the reanalyses

Our offline transport model is used as a tool to perform a fair comparison of advective transport in each reanalysis data-set, using their native vertical grids but a common, low-resolution latitude-longitude grid. It requires on input the surface pressure and horizontal velocity on a so-called Arakawa C-grid, i.e., the zonal wind $u$ must be staggered in longitude and the meridional

wind $v$ must be staggered in latitude. As indicated by its name, the FFSL algorithm evaluates internally the corresponding mass fluxes and derives the vertical winds ($w$) from mass conservation. Hence the reanalysis datasets must be carefully pre-processed from spectral or high-resolution gridded fields to the low-resolution C-grid. We have paid special attention to this pre-processing of the reanalyses to make sure that the different types of wind fields are expressed in a consistent manner for our transport algorithm.

Due to its assimilation procedure, the early ERA-40 reanalysis contained large dynamical imbalances which deteriorated the Brewer-Dobson circulation through excessive upward motion in the tropics and excessive transport from the tropics to the mid-latitudes (Meijer et al., 2004; Monge-Sanz et al., 2007). Pawson et al. (2007) described a similar issue with MERRA and proposed to use time-averaged input wind fields in order to remove these imbalances, but this approach is available only for MERRA and MERRA-2. Here we use a pre-processor, originally developed by Segers et al. (2002) for analyses made at

ECMWF, to filter out such dynamical imbalances. Using the primitive variables of spectral dynamical cores, i.e., the vorticity and divergence expressed on a spherical harmonics basis, this pre-processor evaluates the zonal and meridional winds on a regular latitude-longitude grid while correcting for the small inconsistencies in the pressure tendency compared with the divergence fields. This correction ensures consistent mass fields even in the presence of spurious surface pressure increments which may be caused by data assimilation.

Our pre-processing for the five reanalysis systems is based on this algorithm, with a preliminary derivation of the spherical harmonics coefficients of vorticity, divergence and surface pressure for the reanalyses other than ERA-I. In all cases these spectral coefficients are truncated at the wavelength number 47 to avoid aliasing on the 2°x2.5° target grid (Krishnamurti et al., 2006, section 7.4).

### 2.4 Comparison of Age of Air output with another model

Figure 1 compares the results of the BASCOE CTM driven by ERA-I with those by a reference Eulerian model, using the standard layout of zonal means at 20 km height and at equatorial, middle and polar latitudes (e.g., Waugh and Hall, 2002). Both models transport idealized tracers which increase linearly at the surface and are driven during 20 years by repeating reanalyses of the year 2000. The reference model is TOMCAT, driven by ERA-I analyses with 6-hourly updates. At 20 km height we use the results published by Dee et al. (2011, Fig. 28) while the vertical profiles are those published by Monge-Sanz

et al. (2012, Fig. 1). Some observational context is provided with in-situ observations of $SF_6$ and $CO_2$ (Hall et al., 1999). As done in recent intercomparisons of climate models (Neu et al., 2010; Chipperfield et al., 2014), the AoA value at the equatorial tropopause has been subtracted from the fields in order to exclude the transit time from the surface to the tropopause.



Very good agreement is obtained between TOMCAT and BASCOE CTM. At 20 km height the results are nearly identical except in the southern hemisphere where TOMCAT delivers a slightly weaker latitudinal gradient, resulting in a difference of around 0.5 years above the South Pole between both models. All three vertical profiles show that TOMCAT delivers slightly weaker vertical gradients in the lower stratosphere than the BASCOE CTM. This results in younger mid-stratospheric AoA by

TOMCAT, but here also the largest difference does not exceed 0.5 years (latitude 5°S, height 45 km).

## 3    Intercomparison of AoA values

Time-varying distributions of AoA were derived from each reanalysis for the whole period 1980-2015. The initial conditions were obtained from twenty-year spin-up runs simulating the 1960-1980 period with repeating reanalyses of the year 1980. The importance of the initialization procedure was evaluated with an alternative set of transport experiments starting in 1981 from

forty-year spin-up runs driven by repeating reanalyses of the year 1981. While the initial AoA could be significantly different depending on the initialization procedure (up to 15% difference in 1981 in the case of CFSR), by 1989 these differences were smaller than 1% at all latitudes and pressure levels for each reanalysis (not shown). Hence the five AoA datasets are studied only over the period 1989-2015.

For the sake of convenience the results of each simulation will be designated by its driving reanalysis but the reader is

reminded that all results presented here are obtained indirectly through an offline and kinematic transport model. The outcome of the intercomparison could have been different if the AoA had been computed directly in each reanalysis system.

### 3.1    Mean distribution in 2002–2007

The AoA distributions are first averaged over the period 2002–2007 in order to remove seasonal and quasi-biennal oscillations and also to allow comparisons with the distribution most recently derived from MIPAS observations of $SF_6$ (Kovács et al.,

2017).

The global distribution of AoA is first compared with latitude-pressure cross-sections. ERA-I is taken as reference because it delivers intermediate values and has been used in AoA studies with several other CTMs (see, e.g., Diallo et al., 2012; Monge-Sanz et al., 2012; Konopka et al., 2015). Figure 2 shows the latitude-height cross-sections of AoA for the period 2002–2007, with a noticeable hemispheric asymmetry: as expected, the latitudinal gradient is significantly stronger in the southern mid-

latitudes and polar regions than in the northern hemisphere, and old air masses reach much lower altitudes above the Antarctic than above the Arctic (e.g., the 5-year isoline starts at 50 hPa above the South Pole and ends at 20 hPa above the North Pole). This is qualitatively confirmed by AoA derived from MIPAS observations of $SF_6$ for the same period (Kovács et al., 2017, Fig. 7d).

The four other reanalyses deliver quite different distributions of AoA (Fig. 3). One can distinguish JRA-55 and CFSR as the

"younger reanalyses" with AoA not exceeding 5 years in the polar upper stratosphere; MERRA as the "older reanalysis" with maximum AoA values as large as 6.5 years; and ERA-I with intermediate results (5.8 years in the same regions). MERRA-2 is a special case, with upper stratospheric values similar to those reached by ERA-I but quite different latitudinal gradients. The





hemispheric asymmetry is more pronounced with ERA-I than with any other reanalysis, e.g., the 3 and 4-year isolines (JRA-55 and CFSR, respectively) or the 5-year isoline (MERRA-2 and MERRA) reach nearly the same level above the North Pole than above the South Pole. MERRA-2 stands out in the middle stratosphere with nearly vertical isolines, i.e., very small vertical gradients which are not supported by MIPAS observations (Haenel et al., 2015; Kovács et al., 2017).

While this qualitative comparison of the AoA distributions points to different gradients in the mid-latitudes and polar regions, the relative differences with respect to ERA-I are largest in the tropical lower stratosphere (bottom row of Fig. 3). Hence we focus on this region and its differences with the midlatitudes. Figure 4 shows the intercomparison of AOA zonal means at 50 hPa, at tropical and northern midlatitudes, and the latitude gradient between these two latitude bands. This layout is inspired by the AoA intercomparisons in GCCMs (Neu et al., 2010; Chipperfield et al., 2014), including the subtraction of the AoA values
at the equatorial tropopause.

       The intercomparison at 50 hPa (Fig. 4a) shows again the important disagreements between the five model simulations. JRA-55 yields the youngest AoA at all latitudes with values ranging from 0.8 years at the equator to 3.6 years at the South Pole, while MERRA and MERRA-2 yield the oldest AoA with 1.6 years at the equator and around 5 years a the South Pole. CFSR and ERA-I yield intermediate results with nearly identical values in the northern extratropics but different latitude gradients
in the tropics and southern hemisphere. The sole simulation to deliver a minimum AoA in the southern tropics is driven by CFSR (at 6°S). In the other simulations this minimum is either exactly at the equator (JRA-55, MERRA) or slightly north of the equator (ERA-I, MERRA-2). In the southern hemisphere CFSR results in AoA nearly as young as JRA-55 while ERA-I reaches larger values which are very close to the observations. Overall, the spread between the five simulations at 50 hPa is larger than the $1\sigma$ observational uncertainties in the tropics, and nearly as large in the extratropics. This spread is as large as
in an intercomparison of 7 GCCMs (Chipperfield et al., 2014, Fig. 2), which is a remarkable result considering the use of reanalyses rather than unconstrained climate models.

       The vertical profiles of AoA (Fig. 4b and 4c) confirms that this large spread and general hierarchy of AoA (youngest with JRA-55, oldest with MERRA and MERRA-2) are found at all stratospheric levels. In the northern midlatitudes, MERRA-2 stands out with vertical gradients which are significantly larger in the lower stratosphere but significantly smaller in the upper
stratosphere than in all other reanalyses. While the intermediate values by ERA-I and CFSR agree well with observations in the tropics, this is not the case in the northern midlatitudes where only MERRA and MERRA-2 deliver AoA as old as the observations.

       The AoA difference between the tropics and mid-latitudes (Fig. 4d) is directly related to the inverse of the tropical upwelling velocity and is independent of quasi-horizontal mixing: a smaller AoA latitudinal gradient indicates faster tropical ascent (Neu
and Plumb, 1999; Linz et al., 2016) Except for MERRA-2, the AoA latitudinal gradients delivered by the four other reanalyses agree much more closely than the AoA profiles themselves, at least during the 2002–2007 period. The spread between the four reanalyses reaches a maximum of 0.2 years at 30 hPa, much tighter than the spread of 0.8 years (after removal of one outlier) in the corresponding intercomparison of 6 GCCMs (Chipperfield et al., 2014, Fig. 3c). While there is good agreement with the observation-derived latitudinal gradient below 60 hPa and at 10 hPa, these four reanalyses significantly underestimate
it at intermediate pressure levels. This indicates an overestimation of the tropical upwelling obtained with ERA-I, CFSR,



JRA-55 and MERRA in the lower stratosphere. MERRA-2 yields an outlying vertical profile of AoA at northern midlatitudes, resulting in a latitudinal gradient which is an outlier as well. MERRA-2 apparently underestimates the tropical upwelling in the lowermost stratosphere (100-60 hPa), agrees well with observations at 50 hPa and joins the results of the four other reanalyses at higher levels.

## 3.2 Time evolution and absence of volcanic impact

The Pinatubo eruption, which started on 15 June 1991, is expected to have had a significant impact on AoA (Muthers et al., 2016; Diallo et al., 2017). The assimilation of satellite radiance measurements by the Advanced Microwave Sounding Unit (AMSU) started in 1998 (on 1 August in ERA-I and JRA-55 and 1 November in CFSR, MERRA and MERRA-2) and was repeatedly shown to have a important influence on their description of the stratospheric dynamics (e.g., Simmons et al., 2014; Kawatani et al., 2016; Long et al., 2017). Hence we repeat the latitudinal gradient diagnostic but for the period 1992–1997, i.e., after the Pinatubo eruption and before the ingestion of AMSU radiances (Fig. 5). The general outcome is the same as during the later period: the tropical ascent is too fast with all reanalyses except with MERRA-2. Yet MERRA-2 provides a better match with the observations during this earlier period, and the four other reanalyses do not agree as closely.

Figure 6 shows the globally averaged time evolution of simulated AoA according to the five reanalyses, from 1989 until 2015 at 50 hPa in the midlatitudes. The results are smoothed with a one-year running mean in order to highlight the long-term trends. The overall hierarchy of ages shown on previous figures for year 2002–2007 holds for the whole 1989-2015 period: MERRA and MERRA-2 deliver the oldest AoA, JRA-55 and CFSR the youngest. While MERRA and MERRA-2 agree well in the southern hemisphere, this is not the case in the northern hemisphere where MERRA-2 starts with much older values. A rapid decrease of MERRA-2 values during the 1990's allows these two datasets to reach better agreement after 1998, i.e., the beginning of AMSU assimilation. The MERRA output in the northern hemisphere delivers seasonal cycles with much larger amplitudes than those obtained from all other reanalyses.

The Pinatubo eruption does not appear to have any impact of the simulated AoA at 50 hPa except with MERRA-2 which shows an increase in the southern midlatitudes. This absence of volcanic impact in the other reanalyses is even more evident in a deseasonalized time series of the extra-polar lower stratosphere (Fig. 7). This diagnostic is inspired by Diallo et al. (2017) who showed a significant impact of the Pinatubo eruption on AoA using ERA-I and JRA-55 but with another offline transport model. Since our results apparently contradict this finding, this issue will be further discussed in section 5.

Figure 8 displays time series of AoA in the middle stratosphere (mean values between 30 hPa and 5 hPa). The intercomparison in the southern hemisphere shows large disagreements between the long-term trends among the five reanalyses. MERRA and MERRA-2 values decrease quickly until 1995 and increase after 2007 while ERA-I values follow an opposite pattern. The long-term evolution of AoA in this region is completely different with JRA-55 (gradual decrease during until 2002 followed by a stabilization) and differs yet again with CFSR (no apparent trend before 1997 and rapid increase during 1997-2003).

The thin lines allow a qualitative comparison of faster variations. The seasonal signal dominates in all cases, with similar phases: AoA is oldest in fall and youngest in spring. The seasonal amplitudes are very dependent on the input reanalysis and on the considered year, so their detailed analysis is deferred to the next section. Yet we note already that some reanalyses exhibit





a strong modulation of the seasonal cycle by the Quasi-Biennal Oscillation (QBO; for a general review see Baldwin et al., 2001) while others do not. This can be seen very clearly during the period 2005-2009 when the seasonal amplitudes of AoA by ERA-I and MERRA are approximately twice smaller during the easterly phase of the QBO (i.e. in 2006 and 2008) than during the westerly phase (i.e. in 2005,2007,2009). This modulation of the seasonal variations is weaker in the MERRA-2 and

JRA-55 datasets and absent from the CFSR dataset.

The right plot in Fig. 8 compares the model results in the northern hemisphere with balloon observations collected since the 1970's (Engel et al., 2017), where the outer error bars denote the overall uncertainty of the mean-age value including an assessment of the representativeness of a single profile (Engel et al., 2009). The spread between the five simulations is as large as the observational uncertainties, highlighting again the importance of the disagreements between the five reanalyses. ERA-I

delivers a weakly positive trend over the period 1989-2015, apparently in agreement with the balloon observations, but Engel et al. (2009, 2017) showed that the sign of this observational trend is not significant. More importantly, the AoA from the BASCOE simulation with ERA-I does not show any overall trend after 2000, unless one arbitrarily ends the period in 2010. No reanalysis delivers any change larger than half a year over the whole period 1989-2015 except for MERRA-2 which indicates a large decrease of 0.8 years, but this decrease starts from values much larger than the observations and happens mostly before

2000.

## 4   Analysis of temporal variations

We now perform a quantitative investigation of the temporal variations in order to derive the amplitudes of periodic variations and the linear trends of AoA at all latitudes and pressure levels, including their uncertainties.

### 4.1   Methodology

Vigouroux et al. (2015) used a multiple linear regression model to study the trends of ozone total columns and vertical distribution at several ground-based stations. Here we apply the same tool to $A(t)$, the monthly zonal means of AoA as a function of time, latitude and pressure (after interpolation to a constant log-pressure grid with 2km increments). The multiple linear regression model is expressed as:

$$A(t) = A_0 + A_1 \cdot t + S(t) + Q(t) + \epsilon(t) \tag{1}$$

where $t$ is time, $A_0$ is the baseline value, $A_1$ is the annual trend of AoA and $\epsilon(t)$ represents the residuals. The term $S(t)$ describes the seasonal variations in $A(t)$:

$$S(t) = S_1 \cdot \cos(2\pi t/12) + S_2 \cdot \sin(2\pi t/12) + S_3 \cdot \cos(4\pi t/12) + S_4 \cdot \sin(4\pi t/12) \tag{2}$$

where the coefficients $S_1$ to $S_4$ describe the seasonal cycle. The term $Q(t)$ describes the variations due to the QBO and its seasonal modulations:

$$Q(t) = Q10(t) \cdot [Q_1 + Q_2 \cdot \cos(2\pi t/12) + Q_3 \cdot \sin(2\pi t/12) + Q_4 \cdot \cos(4\pi t/12) + Q_5 \cdot \sin(4\pi t/12)]$$
$$+Q30(t) \cdot [Q_6 + Q_7 \cdot \cos(2\pi t/12) + Q_8 \cdot \sin(2\pi t/12) + Q_9 \cdot \cos(4\pi t/12) + Q_{10} \cdot \sin(4\pi t/12)] \tag{3}$$



where the explanatory variables $Q10(t)$ and $Q30(t)$ are the zonal winds observed above Singapore at 10hPa and 30hPa (data from the FU Berlin: http://www.geo.fu-berlin.de/en/met/ag/strat/produkte/qbo/index.html) and $Q_1$ to $Q_{10}$ are the coefficients associated to these two proxies, including their seasonal dependence.

The uncertainties arising from the fit are corrected for auto-correlation in the residuals (Eqs. 3,4 and 6 in Santer et al., 2000). Preliminary tests also included additional terms to account for the El Niño-Southern Oscillation (ENSO), the 11-year solar cycle and volcanic forcings but it was found that these terms do not impact significantly the linear trends nor the amplitudes of seasonal and quasi-biennal oscillations. Hence they were removed from the regression model in order to avoid any over-fitting of the data and to ease the interpretation of the results.

An important goal of this analysis is the determination of linear trends. As seen in Fig. 6 and 8, such trends depend closely on the considered time period. Hence the regression model was applied not only to the whole simulation period (1989-2015) but also to an "'early period'" (1989-2001) and a "'recent period'" (2002-2015) which starts after the assimilation of AMSU and on the same year as the MIPAS mission (Stiller et al., 2008, 2012).

## 4.2 Amplitudes of the seasonal cycle and Quasi-Biennal Oscillation

The amplitude of the seasonal variations is approximated by the difference between the maximum and minimum values reached by the term $S(t)$ in the linear regression model. Figure 9 shows the dependence of this approximated amplitude with respect to pressure in the polar regions and midlatitudes. The vertical structure agrees broadly across all five reanalyses and in all four regions with maximum amplitudes in the lower stratosphere (around 100 hPa), except above the South Pole where the amplitudes are maximum in the middle stratosphere (10–30 hPa). The ERA-I results agree with an earlier modeling study which also found that seasonal amplitudes are maximum in the lower stratospheric levels of the midlatitudes and North Pole (Diallo et al., 2012, Fig. 9).

MERRA and MERRA-2 stand out with larger amplitudes in the lower stratosphere, resulting above the South Pole in a second maximum which is not found by the three other reanalyses. One may argue that these larger seasonal amplitudes are a direct consequence of the larger annual means (see Fig. 3) but this is not supported by the agreement of JRA-55 and (after 2001) CFSR with ERA-I despite their significantly younger annual means. MERRA and MERRA-2 also deliver very different amplitudes depending on the period used for the analysis, with seasonal amplitudes significantly larger during 1985-2001 than for the recent period. An opposite dependence is noted with CFSR above the South Pole, where after 2001 the seasonal amplitudes became much larger and closer to those delivered by the other reanalyses. The considered period has a comparatively much smaller impact on the seasonal amplitudes in JRA-55 and ERA-I.

We now investigate the differences in the QBO among all reanalyses. Kawatani et al. (2016) have compared the monthly-mean zonal wind in the equatorial stratosphere between reanalyses and found that their degree of disagreement depends on latitude, longitude, height, and the phase of the QBO. They also noted a tendency for the agreement to be best near the longitude of Singapore, suggesting that the Singapore observations act as a strong constraint on all the reanalyses.

Here we perform an intercomparison of the amplitude of the QBO signal (in years) in each reanalysis. We approximate it again as the difference between the maximum and minimum values reached by the term $Q(t)$ in the linear regression model.



Our results for ERA-I show that the QBO amplitude is largest in the subtropics around 30 hPa (not shown), which confirms again the results of Diallo et al. (2012). Fig. 10 compares the results at this pressure level. Except for CFSR, the latitudinal dependence is similar in all reanalyses: the approximated QBO amplitude reaches maximum values around 15 degrees latitude in both hemispheres and presents a marked minimum around the equator. Outside of the equatorial region, the QBO amplitudes

by JRA-55 are significantly smaller than by ERA-I, MERRA and MERRA-2. The amplitudes computed from CFSR show no clear structure in the southern hemisphere and reach unexpectedly large values at the North Pole.

Overall, the reanalyses have large (up to a factor of 2) disagreements with respect to the seasonal amplitudes in the polar regions and QBO amplitudes in the Tropics. The results for MERRA, MERRA-2 and CFSR have strong (up to 50%) dependencies on the considered time period. Such disagreements and dependencies could not be expected from inspection of the

10 native dynamic variables contained in the reanalyses, neither for the seasonal amplitudes of polar stratospheric temperatures nor for the QBO zonal wind anomalies at 10 hPa (see, e.g., Long et al., 2017, Fig. 11 and 10, respectively).

### 4.3 Linear trends

Current research on AoA trends has largely focused on a dipole-like latitudinal structure for the period 2002-2012, which was first derived from satellite observation of $SF_6$ by the MIPAS instrument (Stiller et al., 2012). This structure of trends shows

AoA decreasing in the Southern Hemisphere but unexpectedly increasing in the Northern Hemisphere and could be due to a southward shift of subtropical transport barriers (Stiller et al., 2017). The ERA-I reanalysis supports a dipole-like latitudinal structure of AoA trends, at least since 2002. Haenel et al. (2015, hereafter H2015) derived AoA trends from the distribution of $SF_6$ over the period 2002-2012, using MIPAS observations and a CCM nudged towards ERA-I below 1 hPa, and found a good agreement for the signs, range and latitudinal structure of AoA trends (see Fig. 6 and 10 in H2015). Here we aim to verify our

methodology through a comparison of our results with H2015, to check the consistency of AoA trends derived from the four other reanalyses, and to explore the latitudinal structure of AoA trends for periods starting earlier than 2002.

The linear trend is represented by $A_1$ in the multiple regression linear model (Eq. 1). It is expressed in years per decade (yr dec$^{-1}$) and is deemed significant at a given grid point if its absolute value is larger than its standard error. Figure 11 presents the ERA-I trends during the period 2002-2012 in order to compare with H2015. In the polar regions, H2015 showed large and

25 positive trends while they are insignificant according to our model (Fig. 11). This disagreement can be attributed to different approaches: here we study the true age of air using a theoretical tracer with no losses, while H2015 evaluated the apparent mean age of air taking into account the mesospheric sink of $SF_6$ which has the largest impact in the polar regions (Reddmann et al., 2001). Outside of the polar regions, Figure 11 shows good agreement with both observational and modeling results in H2015, including with respect to the significance of the trends: in the 30-60 hPa (approx. 25-20 km) layer the trends are significant at

30 all extra-tropical latitudes, negative in the southern hemisphere and positive in the northern hemisphere.

Figure 12 compares the latitude-pressure distributions of AoA trends across all five reanalyses and for the early (1989-2001), recent (2002-2015) and overall periods (1989-2015). It is important to note that the trends over the early and overall periods should be considered with caution since there was little data to constrain the stratospheric winds until 1998 (see the discussion in the next section). The AoA trends derived from ERA-I wind fields during the early period (Fig. 12, upper left)





show unexpected growth in both hemispheres, except in the northern lowermost stratosphere. During the recent period, the dipole structure derived from ERA-I (Fig. 12, upper middle) is similar but much less clear than over the slightly shorter period 2002-2012 (Fig. 11), with weaker increases in the northern hemisphere which remain significant only in the polar lower stratosphere. The extension of this trend analysis for the overall period (Fig. 12, upper right) shows a dipole structure with

negative but mostly insignificant trends in the southern hemisphere; positive trends in the northern middle stratosphere which mostly corresponds to the region with positive trends during the 1989-2001 period; and significantly negative trends in the lowermost stratosphere at all extra-polar latitudes. Diallo et al. (2012) used a diabatic transport model driven by ERA-I and found for the period 1989-2010 negative AoA trends in the lower stratosphere and positive trends in the mid-stratosphere, suggesting that the shallow and deep Brewer-Dobson circulations may evolve in opposite directions.

Our transport model, using only wind fields and surface pressure from ERA-I, confirms this finding.

Comparing the results obtained with ERA-I with those from other reanalyses, one notes immediately general agreement between ERA-I and CFSR on one hand (Fig. 12, first and second row) and opposite trends in JRA-55, MERRA and MERRA-2(third to fifth row). The agreement between multidecadal trends in ERA-I and CFSR may be related to their closeness in AoA distribution and spatial gradients (section 3.1). For all reanalyses except ERA-I, the trends for the overall period (1989-2015:

Fig. 12, right column) appear dominated by the results from the early period which are subject to caution.

To summarize, the signs of the trends depend strongly on the input reanalysis and on the considered period with values above 10 hPa varying between approximately -0.4 and 0.4 years per decade. JRA-55, MERRA and MERRA-2 indicate an AoA increasing globally over 2002-2015, except in the lowermost stratosphere; while ERA-I and CFSR indicate exactly the opposite (Fig. 12, middle column). These trends are significant only in specific regions of the stratosphere, and the regions of

significance vary depending on the considered reanalysis. ERA-I stands out as the only reanalysis yielding a dipole structure of AoA trends for the period 2002-2015, although one may note that in the lower stratosphere, the AoA growth derived for this period from MERRA and MERRA-2 (Fig. 12, middle column, fourth and fifth row) is faster in the northern hemisphere than in the southern hemisphere. The most striking result, in this intercomparison of AoA trends for different periods, is the reversal of trends between the early (1989-2001) and recent (2002-2015) periods. This reversal is found for all five reanalyses and in

all regions of the stratosphere.

## 5  Discussion and outlook

This intercomparison reveals large disagreements between the AoA derived from the five reanalyses, both with respect to their values and their linear trends. The spread of AoA at 50 hPa (Fig. 4a) is as large as in an intercomparison of CCM (Chipperfield et al., 2014). An intercomparison of AoA trends during the twenty-first century among 6 CCM shows negative trends in the

whole middle atmosphere with no large hemispheric asymmetry (Butchart et al., 2010) while our results for 1989–2015 show different signs depending on the reanalysis and the stratospheric region. Since these results call for further research, we propose here a summary overview of the possible causes for these disagreements and some venues to attempt their identification.




Many intercomparisons of reanalyses have focused on the instantaneous values or long-term evolution of direct output fields such as temperature or zonal winds (Simmons et al., 2014; Lawrence et al., 2015; Long et al., 2017; Kozubek et al., 2017). These intercomparisons do not find large discrepancies, especially after the introduction of new satellite instruments around year 2000. The large disagreements obtained here may look surprising unless one considers the lack of wind observations available for

assimilation in the tropics, high latitudes and stratosphere (Baker et al., 2014). This deficiency of wind information may explain the divergences between trajectories obtained with different reanalyses in the lower stratosphere, e.g., in the equatorial region during some phases of the QBO (Podglajen et al., 2014) or above the Antarctic during the vortex break-up season (Hoffmann et al., 2017). Such divergent trajectories could have a significant cumulative impact on the mean Age of Air because it is a time-integrated diagnostic spanning several years.

Since the wind fields are weakly constrained, the causes for the disagreements found here may lie in the differences between the underlying models which were summarized recently in the context of S-RIP (Fujiwara et al., 2017). Let us first look at vertical resolution, which has an important impact on the modeling of lower stratospheric dynamics (Richter et al., 2014). In the lower stratosphere, the vertical resolution of CFSR is finest while the resolution of and ERA-I and JRA-55 is the coarsest, with the resolution of MERRA and MERRA-2 in between (Fujiwara et al., 2017). This has no clear impact on AoA since CFSR

and JRA-55 deliver the youngest AoA while the MERRA and MERRA-2 deliver the oldest, with ERA-I results in between. Hence one cannot establish a simple link between vertical resolution and AoA in this intercomparison.

The present intercomparison cannot establish the impact of different horizontal resolutions because it uses a common horizontal grid with a coarse resolution of $2° \times 2.5°$ (see sections 2.1 and 2.3). For example, the intercomparison of AoA distributions (section 3.1) showed that JRA-55 and CFSR yield the weakest latitudinal gradients despite their horizontal grid spacing

which is finest among the five reanalyses studied here (see Fujiwara et al., 2017, table 2). Another intercomparison could yield different results if it uses the wind fields in each reanalysis at its original resolution – but this could lead to difficulties in the handling of horizontal diffusion (Jablonowski and Williamson, 2011).

Different parametrizations of gravity wave drag are another possible modeling cause for the disagreements in AoA. ERA-I, JRA-55 and CFSR all neglect non-orographic gravity wave drag (except for CFSv2, i.e., CFSR after 2010) and each uses its

own parametrization of orographic gravity wave drag. MERRA and MERRA-2 on the other hand use the same parametrization for orographic gravity wave drag (McFarlane, 1987) and both take non-orographic gravity wave drag into account. While this may be a coincidence, MERRA and MERRA-2 happen to provide much older AoA than the three other reanalyses.

Miyazaki et al. (2016) compared the mean-meridional circulations and also the mixing strengths in six reanalyses – including ERA-I and JRA-55 – and also found significant disagreements. Their diagnostics are closely related to AoA since a faster mean-

meridional circulation evidently leads to younger AoA and increased mixing corresponds mostly to additional aging of air due to recirculation from the extra-tropics to the Tropics (Garny et al., 2014). For example, the disagreements of linear trends for 1989-2015 (right column in Fig. 12) confirm the finding that ERA-I and JRA-55 have opposite linear trends of tropical upward mass flux for the period 1979-2012, with fluxes increasing at all levels in JRA while in ERA-I they increase only in a shallow layer of the lower troposphere but decrease in the middle stratosphere (Miyazaki et al., 2016, Fig. 11).



MERRA-2 stands out with outlying AoA values during the 1990's. A connection is plausible with its difficulties to represent correctly the QBO before 1995 (Kawatani et al., 2016; Coy et al., 2016). Interestingly, Gelaro et al. (2017) noted on that same year a marked decrease in temperature near 1hPa and associated it with a change in assimilated radiance data. Gelaro et al. (2017) describe three features which are absent from the other reanalysis systems and could also play a role in the description

of middle atmosphere dynamics in MERRA-2, contributing to its outlying AoA. With respect to assimilated observations, MERRA-2 is the only reanalysis to assimilate Aura-MLS temperatures, from 2004 onwards and above 5 hPa. While this has an important impact on temperatures in the upper stratosphere and lower mesosphere, it does not seem to have an impact on the AoA time series in the middle stratosphere (Fig. 8) and cannot explain the large values obtained during the 1990's. With respect to forward model forcings, MERRA-2 is the only reanalysis which includes a large source of non-orographic gravity

wave drag in the tropics (Molod et al., 2015) and realistic aerosol optical depths. This last feature most probably explains the sensitivity of the MERRA-2 AoA at 50 hPa to the Pinatubo eruption, which cannot be seen with any other reanalysis (Fig. 6).

Yet the impact of the Pinatubo eruption on MERRA-2 AoA at 50 hPa cannot be seen in the northern midlatitudes, and in the southern midlatitudes it is not larger than the amplitude of seasonal variations. In section 3.2 we could not find any influence of volcanic aerosols at the global scale (Fig. 7), contrarily to recent results obtained by Diallo et al. (2017) using the Chemical

Lagrangian Model of the Stratosphere (CLaMS) driven by ERA-I and JRA-55. While CLaMS is a Lagrangian transport model and BASCOE CTM a Eulerian transport model, we believe that these conflicting results are better explained by the different approaches with respect to vertical transport: BASCOE CTM is a kinematic model (see section 2.1) while CLamS is a diabatic transport model, hence also driven by the heating rates from the reanalysis forecast models (Ploeger et al., 2010, 2015b).

Wright and Fueglistaler (2013) have shown that the heat budgets differ significantly in the Tropical Tropopause layer among

20 the reanalyses, with substantial implications for representations of transport and mixing in this region. Abalos et al. (2015) evaluated the vertical component of the advective BDC in ERA-I, MERRA and JRA-55 and found substantial differences between direct (i.e. kinematic) estimates and indirect estimates derived from the thermodynamic balance (i.e. using diabatic heating rates). These intercomparisons of dynamical diagnostics highlight the need for another intercomparison of AoA using a diabatic transport model, because this approach would also reflect the differences between the diabatic heat budgets of each

25 reanalysis - including the temperature increments from the assimilation of temperature radiances (Diallo et al., 2017).

Future work will also involve the disentangling of the contributions to AoA of residual circulation, mixing on resolved scales and mixing on unresolved scales (i.e. diffusion) as recently performed with ERA-I (Ploeger et al., 2015a; Dietmüller et al., 2017) and quantitative comparisons with observational data-sets, using both MIPAS observations of $SF_6$ (Stiller et al., 2012; Haenel et al., 2015) and balloon observations of $SF_6$ and $CO_2$ (Ray et al., 2014). Comparisons with long-term records of

30 other long-lived tracers will provide further insight at multidecadal scales. A recent study by Douglass et al. (2017) explained that the relationship between AoA and the fractional release of such tracers is a stronger test of the realism of simulated transport than the simple comparisons of mean age distributions. This approach seems very promising not only in the context of S-RIP but also for observation-based evaluations of stratospheric transport in global circulation-chemistry models.





## 6   Summary and conclusions

We have developed a pre-processor to feed a Eulerian and kinematic transport model with any of the available global reanalysis datasets. This has allowed us to compute the mean Age of Air (AoA) in the stratosphere and its evolution from 1985 to 2015, according to five modern reanalyses: ERA-Interim, JRA-55, MERRA, MERRA-2 and CFSR. Our results compare well with those published previously using other transport models driven by ERA-Interim and MERRA-2.

The five reanalyses deliver very different and diverse results. In the middle and upper stratosphere, MERRA yields the oldest AoA (~5-6 years at mid-latitudes) and JRA-55 the youngest one (~3.5 years). MERRA-2 provides a different distribution of latitudinal and vertical AoA gradients than any other reanalysis, with near-zero vertical gradients in the middle stratosphere which do not seem supported by observations. CFSR and ERA-I give the most similar AoA distributions, with the latter providing stronger gradients vertically in the middle stratosphere and latitudinally in the southern hemisphere. The relative differences between ERA-I and the four other reanalyses are largest in the lower tropical stratosphere. Tropical ascent rates have been compared through the difference between AoA in the northern mid-latitudes and in the tropics, showing good agreements between all reanalyses except for MERRA-2 and an overestimation of the upwelling in the tropical lower stratosphere.

The time variations of AoA were studied first through a qualitative analysis of raw time series in the mid-latitudes, then through a fit with a multiple linear regression model. While the linear trends vary considerably depending on the considered period (2002-2012, 2002-2015 or 1985-2015), the general hierarchy of "older" (MERRA, MERRA-2) and "younger" (JRA-55, CFRS) reanalyses holds during the whole 1985-2015 period, with ERA-I keeping intermediate AoA values. The MERRA-2 results stand out again, with an exceptionally large initial AoA in the northern hemisphere which quickly decreases during the 1990's to reach values similar to those in MERRA. A comparison was performed with a time series of balloon observations realized since the 1970's in the northern mid-latitudes where the uncertainties include an evaluation of the sampling error (Engel et al., 2017). The spread between the five simulations is as large as the observational uncertainties, highlighting again the importance of the disagreements between the five reanalyses. The AoA using ERA-I does not show any overall trend after 2000, unless one arbitrarily ends the period in 2010.

The amplitudes of seasonal variations in the lower stratosphere are larger in MERRA and MERRA-2 than in the three other reanalyses. The seasonal amplitudes of MERRA-2 decrease significantly during the 2002-2015 period but at the North Pole they remain 50% larger than those of the three other reanalyses. The latitudinal dependence of QBO amplitudes is similar in all five reanalyses except for CFSR which show no clear structure in the southern hemisphere.

The linear trends of ERA-I AoA confirm again the dipole structure of the latitude-height distribution of AoA trends as derived from MIPAS observations of $SF_6$ for the 2002-2012 period (Haenel et al., 2015), with a decrease in the southern hemisphere and an increase in the northern lower stratosphere which is significant and not expected from climate model simulations. Yet the trends derived from ERA-I are shown to closely depend on the considered period. When it is extended to 2002-2015, the positive trends in the northern hemisphere become mostly insignificant and the dipole structure becomes much less pronounced. A further extension to 1989-2015 shows positive trends which become significant again in the northern middle stratosphere, but the negative trends in the southern middle stratosphere become insignificant. For all five reanalyses the trends over the early





period (1989-2001) have opposite signs than over the recent period (2002-2015). Looking only at the recent period which is better constrained by observations, the main outcome is again large disagreements between the reanalyses: JRA-55, MERRA and MERRA-2 provide increasing AoA in the middle stratosphere while CFSR provides a decreasing but mostly insignificant trend. Independently of the considered period, no reanalysis other than ERA-I finds any dipole structure in the latitude-height

distribution of AoA trends.

No obvious cause could be found for these disagreements. The parametrization of non-orographic gravity wave drag in the underlying dynamical model deserves further investigation, especially in the case of MERRA-2 which has difficulties to represent correctly the QBO before 1995. No global impact of the Pinatubo eruption can be found in our simulations of AoA, contrarily to a recent study which used ERA-I and JRA-55 to drive a diabatic transport model. This highlights the need to

repeat the present intercomparison with diabatic transport models because they would reflect directly the significant differences between the heating rates in the reanalyses (Wright and Fueglistaler, 2013). Future work will also focus on quantitative comparisons with AoA derived from MIPAS observations of $SF_6$; comparisons with the long-term records of other long-lived tracers to provide further insight at multidecadal scales; and disentangling the contributions to AoA of residual circulation, mixing on resolved scales and mixing on unresolved scales.

The main conclusion of this study is the significant diversity in the distribution of mean AoA which we obtain with our transport model, depending on the input reanalysis. This casts doubt on our ability to model accurately the time necessary for variations of greenhouse or ozone-depleting species to propagate from the troposphere to the stratosphere. We have also found large disagreements between the five reanalyses with respect to the long-term trends of age of air. This suggests that with our type of offline transport model, the wind fields in modern reanalyses are not sufficiently constrained by observations to evaluate

the actual changes of stratospheric circulation. Yet this conclusion should not be hastily extended to other types of transport models which also use the reanalyses of temperature and heating rates.

*Code and data availability.*  The monthly zonal averages of AoA, as delivered by the BASCOE CTM experiments driven by the five input reanalyses, are distributed as an online supplement to this article. The source code of the BASCOE CTM, including its tools to pre-process the reanalyses, is available by email request to the corresponding author. The ERA-Interim reanalysis (Dee et al., 2011) is provided by

the ECMWF, see http://www.ecmwf.int/en/forecasts/datasets. MERRA data (Rienecker et al., 2011) and MERRA-2 data (Gelaro et al., 2017) are provided by the Global Modeling and Assimilation Office at NASA Goddard Space Flight Center through the NASA GES DISC online archive; see https://disc.gsfc.nasa.gov/information/glossary?keywords=merra. The CFSR (Saha et al., 2010) and CFSv2 (Saha et al., 2014) reanalyses data were obtained from NOAA NCEP; see http://cfs.ncep.noaa.gov/. The JRA-55 reanalysis (Kobayashi et al., 2015) was obtained from the NCAR Research Data Archive; see https://rda.ucar.edu/datasets/ds628.0/.

*Acknowledgements.*  We thank the reanalysis centers (ECMWF, NASA GSFC, NOAA NCEP and JMA) for providing their support and data products. We thank Dr. Gabriele Stiller, Dr. Paul Konopka and Dr. Bernard Legras for fruitful discussions during the preliminary steps leading to this study, and Dr. Masatomo Fujiwara for his coordination of the S-RIP. Yves Christophe's contribution was partly supported by the



European Commission project MACC-II under the EU Seventh Research Framework Programme (contract number 283576). Daniele Min-
ganti's contribution was financially supported by the Fonds de la Rechecherce Fondamentale Collective through research project ACCROSS
(convention PDRT.0040.16). Emmanuel Mahieu is Research Associate with the F.R.S.-FNRS.



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





**Figure 1.** Mean Age of Air (AoA) from two model simulations using idealized tracers advected by ERA-I for fixed year 2000. Models shown are BASCOE CTM (blue solid lines) and TOMCAT (blue dotted lines). The modeled AoA fields are corrected so that mean age = 0 at Equator, 15 km. Upper left: values at 20km height; upper right: vertical profiles at 5°S; lower left: vertical profiles at 40°N; lower right: vertical profiles at 65°N. The symbols represent in situ observations collected during the 1990's (see Hall et al., 1999; Waugh and Hall, 2002, for details). The legend in the upper left panel applies to all four panels.





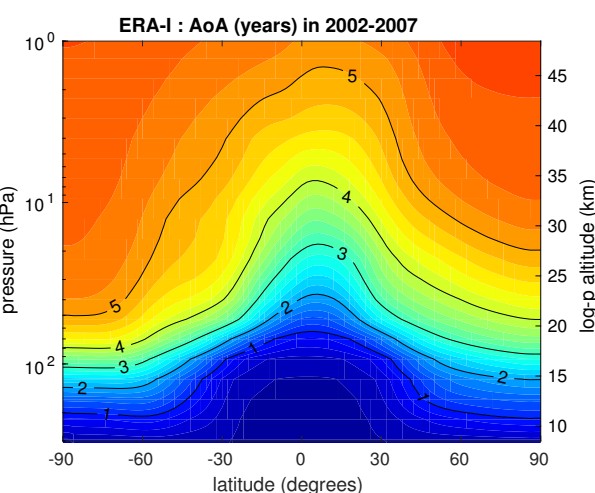

**Figure 2.** Latitude-pressure distribution of mean AoA in 2002–2007 from the BASCOE simulation driven by ERA-I. Blue colors indicate relatively small values, orange and red colors indicate relatively large values.





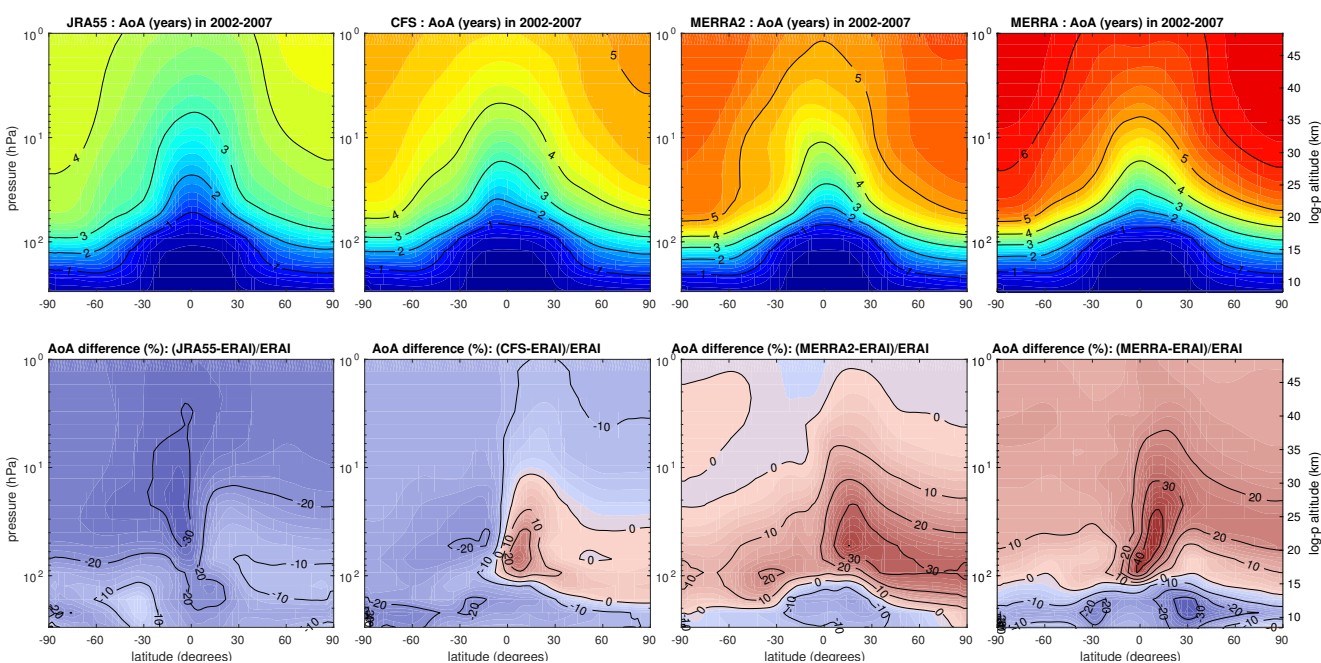

**Figure 3.** Latitude-pressure distribution of AoA in 2002–2007 from BASCOE simulations driven by all reanalyses but ERA-I (top row; same color scale as previous figure) and relative difference with respect to the mean AoA by the ERA-I-driven simulation for the same period (bottom row; darker blues indicate more negative differences and darker reds more positive differences). These reanalyses are, from left to right: JRA-55, CFSR, MERRA-2, MERRA.





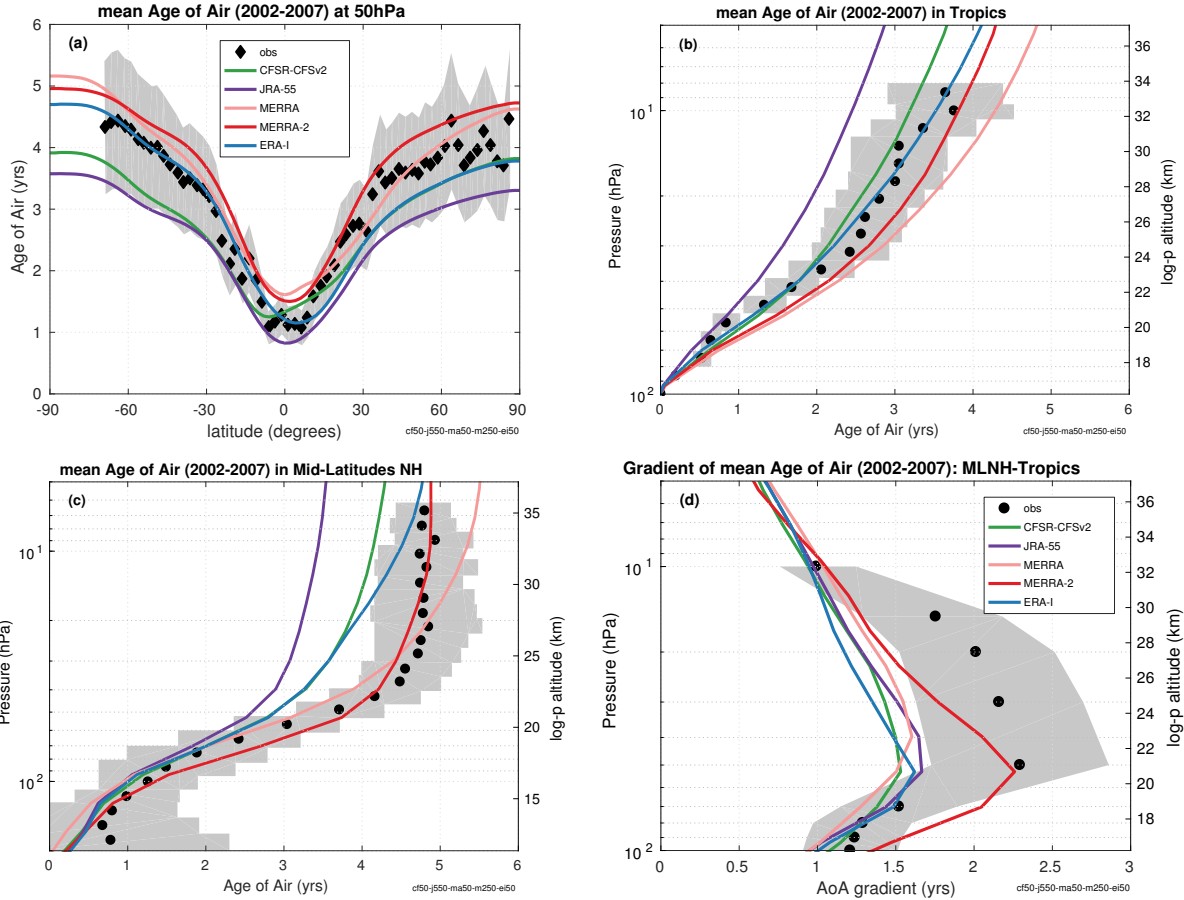

**Figure 4.** AoA in 2002–2007 by the BASCOE TM driven by five reanalyses (solid lines) versus in-situ observations (symbols) with their $1\sigma$ uncertainties (grey shading). The five reanalyses are ERA-I (blue), MERRA-2 (red), MERRA (pink), JRA-55 (purple) and CFSR (green). The modeled AoA fields are corrected so that mean age = 0 at the tropical tropopause (100 hPa). (a) AoA at 50 hPa with aircraft observations of $CO_2$ (Andrews et al., 2001; Neu et al., 2010). (b) AoA in the tropics(10°N–10°S) with aircraft observations (Andrews et al., 2001; Chipperfield et al., 2014). (c) AoA in the northern mid-latitudes (35°N–45°N) with balloon observations (Engel et al., 2009; Chipperfield et al., 2014). (d) AoA gradient between the northern mid-latitudes and tropics (Neu et al., 2010; Chipperfield et al., 2014). The legend in panel (d) applies to panels (b) and (c) as well.





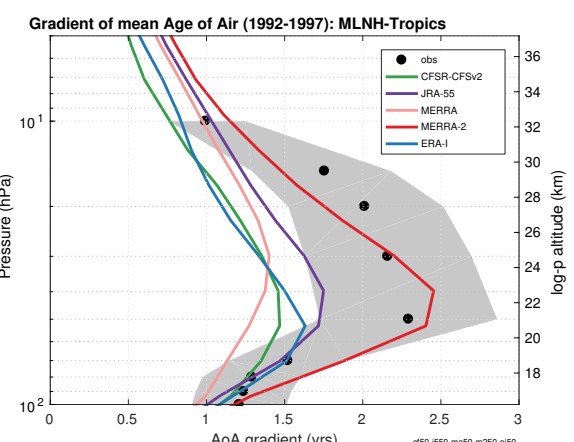

**Figure 5.** Same as figure 4(d) but for the period 1992–1997.




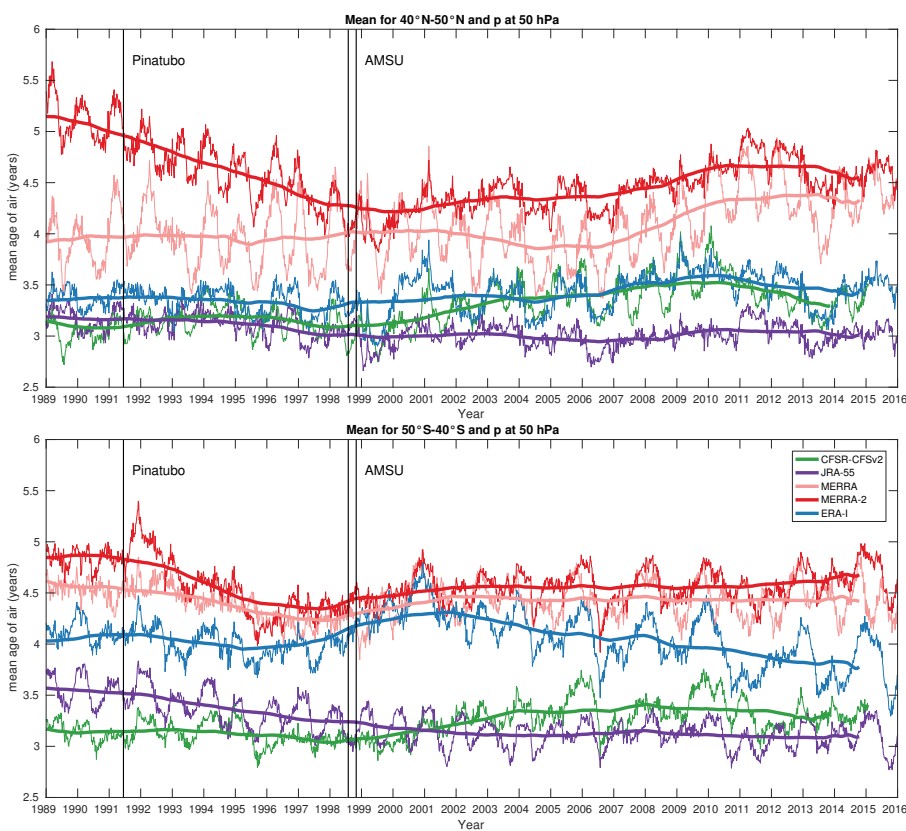

**Figure 6.** Time evolution of AoA interpolated to a pressure of 50 hPa in the northern mid-latitudes (40°N–50°N mean, top) and in the southern mid-latitudes (50°S–40°S mean, bottom). Thin lines show instantaneous model output every 5 days using the five reanalyses with color codes according the the legend shown in the lower panel. Thick lines are smoothed with a one-year running mean. The black vertical lines highlight the start of the Pinatubo eruption and the first assimilation of AMSU (see text).



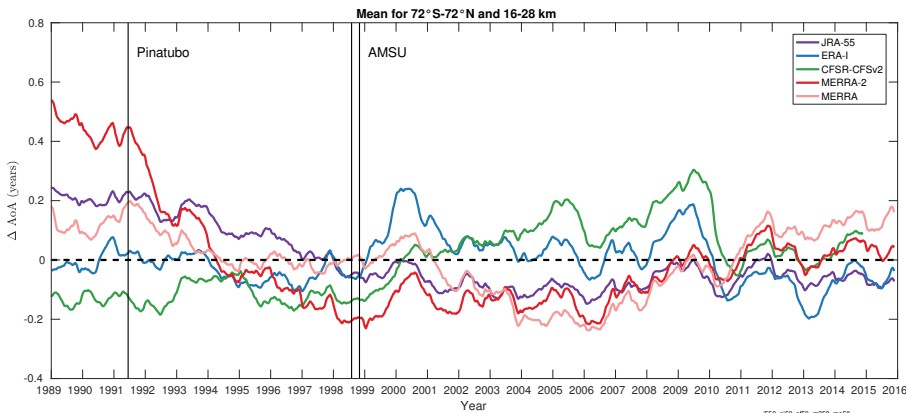

**Figure 7.** Time evolution of the globally averaged (72°S–72°N) anomalies of AoA with respect to their mean (1989–2015) annual cycles, between 16 km and 28 km, using the five reanalyses with same color codes as in previous figure. No impact of the Pinatubo eruption can be seen, contrarily to the results of Diallo et al. (2017, Fig. 1b).





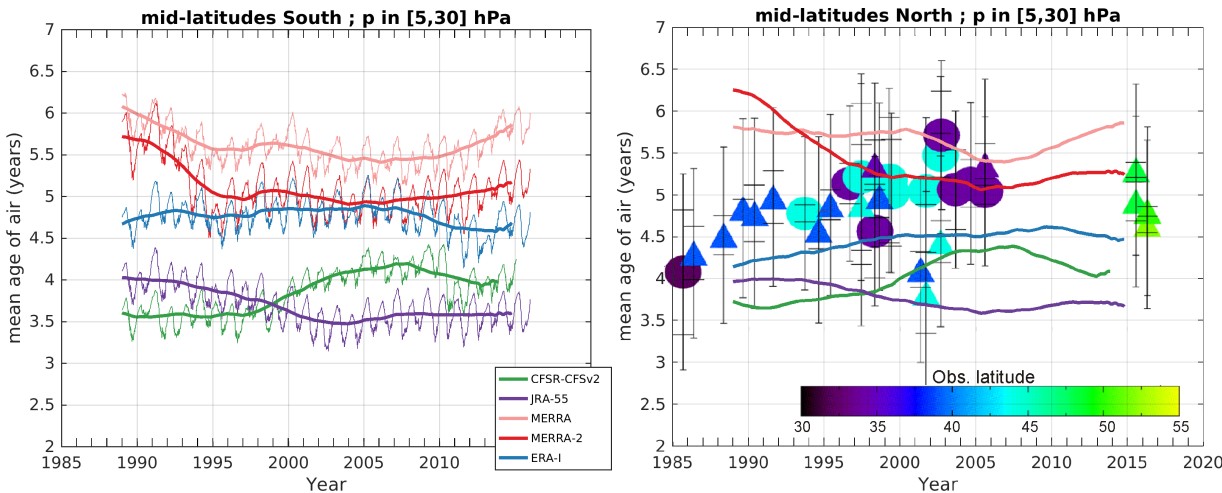

**Figure 8.** Time evolution of AoA averaged from 30 hPa to 5 hPa (approximately 24 to 36 km) in the southern (50°S–40°S, left) and northern mid-latitudes (40°N–50°N, right). Solid lines show model output with color codes according to the legend shown in the left panel. Thin lines (left panel only; omitted from right panel for clarity) show instantaneous model output every 5 days while thick lines are smoothed with a one-year running mean. Northern mid-latitude symbols (right panel) represent values derived from balloon observations of $SF_6$ (circles) and $CO_2$ (triangles) with color code showing the latitude of the measurements and outer error bars including sampling uncertainties (Engel et al., 2017).



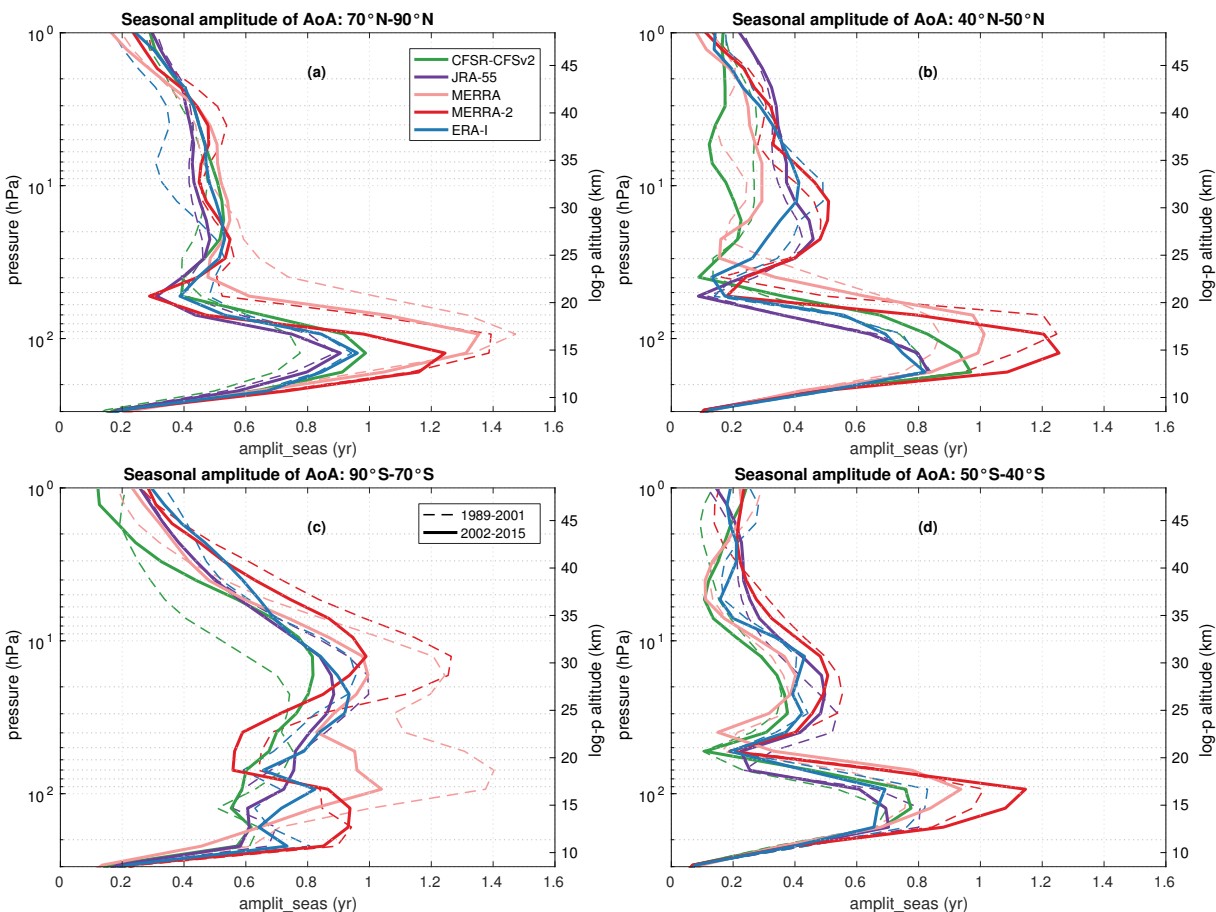

**Figure 9.** Amplitude of the seasonal variation in the linear regression fit of AoA, as a function of pressure and averaged in four latitude bands: a) North Pole, 70°N-90°N; b) Mid-latitudes North, 40°N-50°N; c) South Pole, 90°S-70°S; d) Mid-latitudes South, 50°S-40°S. The regression model was applied separately to AoA results for the periods 1989-2001 (dashed thin lines) and 2002-2015 (thick solid lines). Same color codes as in previous figures: the legends in panels (a) and (c) apply to all four panels.





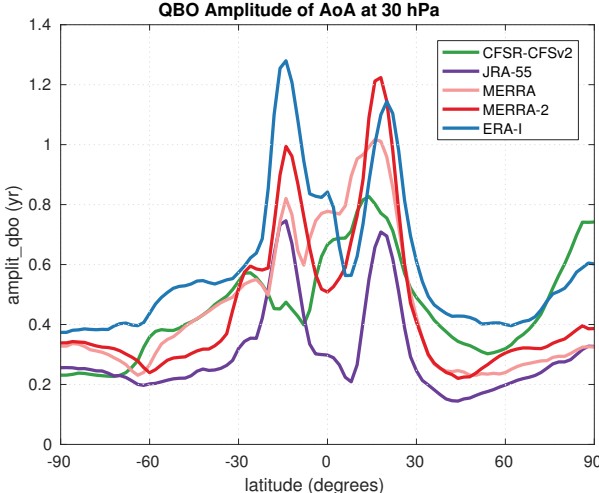

**Figure 10.** Amplitude of the QBO variation in the 2002-2015 linear regression fit of AoA, as a function of latitude at pressure 30 hPa. Same color codes as in previous figures.




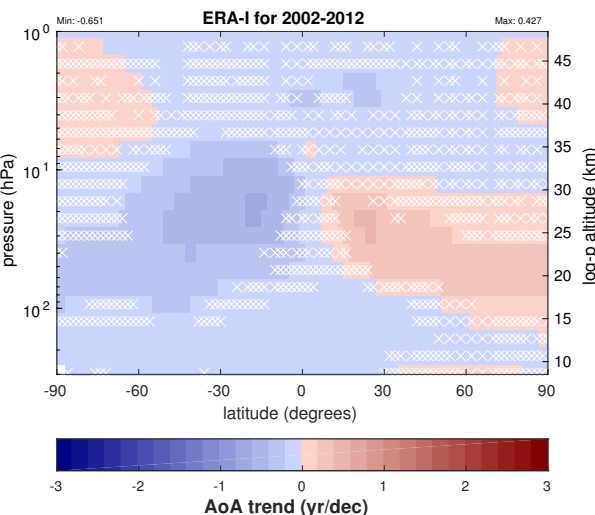

**Figure 11.** Latitude-pressure distribution of AoA trends (years/decade) using the ERA-Interim reanalysis over 2002-2012. White crosses indicate grid points where the sign of the trend is not significant, i.e., its absolute value is smaller than the standard error delivered by the regression analysis. The color scale is the same as in Haenel et al. (2015, Fig. 6 and 10) with darker blues indicating more negative trends and darker reds more positive trends.





**Figure 12.** Latitude-pressure distributions of AoA trends (years/decade) over 1989-2001 (left column), 2002-2015 (middle column) and 1989-2015 (right column) using the five reanalyses (from top to bottom: ERA-I, CFSR, JRA-55, MERRA, MERRA-2). White crosses and colors have the same meaning than in the previous figure, but note the different scale (top of figure).