# Peer review of "Comparison of mean age of air in five reanalyses using the BASCOE transport model"

_Atmospheric Chemistry and Physics, 2018_

## Referee Comment (RC1) · Anonymous Referee #1 · 24 May 2018

The paper presents a comparison of mean age of air (AoA) diagnostics in five modern reanalyses (ERA-Interim, MERRA, MERRA-2 and JRA-55). The AoA is obtained from the kinematic transport model BASCOE. The results reveal large discrepancies in the magnitude and even larger in the trends, which show different sign and spatial structure. The paper is well timed and will make a very useful contribution to the S-RIP Special Issue. It is well written, the methodology is accurately described and the results are clearly presented. I only have the following minor suggestions.

- It is stated that the model input is wind and surface pressure. Is the latter used to convert from model levels to pressure levels? Perhaps this could be said directly at some point.

- Is the sentence in L15-16 (P1) of the abstract needed? It seems redundant.

[Figure]

- L4 P2: "considerably depending on the considered period": consider changing one of the two to avoid repetition.

- L30 P3: explain how these. . .

- L32 P3: each tailored for a different reanalysis dataset.

- L14-15 P4: It could be mentioned why it is chosen not to use the vertical wind component directly.

- L31 P4: "idealized tracer which increases linearly at the surface": throughout the surface or just in the tropics?

- L31-32 P6: "the AoA at the equatorial tropopause has been subtracted from the fields...": did you use the climatological or time-dependent tropopause altitude?

- L9 P8: GCCM: this has not been introduced before, do you mean CCM?

- L14-15 P8: "different latitude gradients between the tropics and ..."

- L14 P9: remove "globally", it is not global but midlatitude average.

- L30 and 31 P9: "different with" should be "different from"

- L11 P10: "not significant": in ERA-Interim or in observations?

- L12 P10: "ERA-I does not show any overall trend after 2000...". Why do you point out these trends after 2000 in ERA-I specifically? Same thing on P16 L22-23.

- L23 P12: standard error for which confidence level?

- L1 P13: remove "unexpected".

- L2 P13: remove "much"

- L7 P13: "Diallo et al. (2012)..." Ploeger et al. (2015a) show AoA trends for the total and later periods.

**[ACPD](https://acpd)**

Interactive
comment

- L10 P13: "using only wind fields": do you mean not using heating rates? Perhaps this should be explicit.

- L26-27 P14: "While this may be a coincidence...": but having more wave drag would imply a faster BDC, so I do not see the point of this sentence.

- L34 P14: "Miyazaki et al. (2016) Fig. 11": The trends in annual mean tropical upwelling for these reanalyses are shown in Abalos et al. (2015) Fig. 11.

- L18 P15: Another difference with CLaMS is that it includes a mixing parameterization.

- Figure 7 caption: "No impact..." This sentence does not belong to the caption but to the main text.

[Figure]

---

## Referee Comment (RC2) · Anonymous Referee #2 · 28 May 2018

This paper examines the distribution of age of air (AoA), its seasonal and quasi-biennial variation, and its long-term trend using result from the BASCOE model driven by output from five atmospheric reanalyses over the period 1980-2015. The principal conclusion is that the simulations of AoA obtained when BASCOE is constrained by different re-analyzed datasets differ substantially from one another. This is not at all unexpected given the differences among the reanalysis models.

The paper is well organized and clearly written, with some exceptions, the main one being that the procedure for computing AoA is not well explained. In particular, it is not clear whether AoA is calculated with respect to a reference level at the tropical tropopause or in the troposphere, and this introduces some ambiguity in the interpretation of the results.

[Figure]

The paper should be suitable for publication in ACP once the following comments are addressed.

Specific Comments (page, line):

(4, 20) "There is no other representation of convection": It is not clear that, in the Tropics, where deep convection can reach the 14-15 km level, this artificial diffusion can simulate vertical transport realistically. But perhaps this does not matter for assessments of AoA in the stratosphere if the base point for AoA calculations is taken to be at or near the tropical tropopause? Please comment (especially since it is not clear how the reference level for computing AoA is chosen). See also comment at (7, 24).

(6, 5) "at the wavelength number 47": "at wavenumber 47" might be better.

(6, 8) "Figure 1 compares the results": I do not believe you have stated how AoA is calculated. In the Introduction (2, 11) you write that "the mean age of air (hereafter AoA) is an evaluation of the time necessary for variations of long-lived ... species to propagate from the troposphere to various regions in the stratosphere". On the other hand, the upper right panel of Fig. 1 suggests that the reference point is at or near 15 km, that is, just below the tropical tropopause. Is that correct? Please be more explicit.

(7, 18) Figure 2: This figure, as well as Figure 3, would benefit from a color bar to indicate the values of the AoA isolines not explicitly labeled.

(7, 24) Figure 3: I am confused by this comparison. One would think that the difference in AoA should vanish at the reference point, assuming the latter is the same for all calculations, but for some of the reanalyses (JRA55, CFS), AoA is younger than in ERAi everywhere. Is this because BASCOE is run on the native grid of each reanalysis, such that there is no common reference point for the AoA calculations? Also, contrary to the BASCOE–TOMS and BASCOE–SD-WACCM comparisons shown in Figure 1, where AoA = 0 in the upper tropical troposphere (Fig. 1, upper right panel), AoA in the simulations shown in Figs. 2 and 3 is not zero at this location. So, where is the

reference point in these simulations? If it is at the surface, then AoA will reflect the effects of transport not just in the stratosphere, but also in the troposphere, including the artificial diffusive transport between the surface and the middle troposphere. Unless I am misunderstanding what you have done here, it seems to me that, if AoA is intended to highlight transport in the stratosphere (e.g., Waugh and Hall, 2002, Sec. 3.1) then the choice of a base point in the troposphere confuses the issue, especially given the use of artificial diffusive transport in the lower troposphere.

(8, 5) "The intercomparison at 50 hPa": You should state explicitly in the text that in these comparisons AoA is "normalized" to be zero at the tropical tropopause (this is only stated in the caption of Figure 4). Otherwise, the reader will wonder, as I did, why the AoA shown in Figures 2-3 are different from the AoA in Figure 4. By the way, a problem with the "normalization" of AoA to zero at 100 hPa is that it gives the impression that AoA above that level is determined only by the stratospheric circulation, when in fact the AoA also contains the effect of transport in the troposphere.

(8, 12) "overall, the spread . . . is larger than the 1-sigma. . .": One wonders how this result would change if AoA were computed with respect to a reference point at 100 hPa.

(8, 26) "The spread between the four reanalyses reaches a maximum of 0.2 years at 30 hPa": Are you referring here to the gradient comparison, Figure 4d? How is this "gradient" calculated? The figure legend refers to "MLNH-Tropics" and shows values in months, not per unit distance, so this is really a difference between the Tropics and midlatitudes of the NH. How are Tropics and NH midlatitudes defined? Note also that the "outlier" behavior of the MERRA-2 simulation is confined to altitudes below 22 km or so, where, interestingly, it agrees better with the data than any other simulation.

(8, 30) "MERRA-2 yields an outlying vertical profile of AoA at northern midlatitudes": True with respect to the other reanalyses except for MERRA (Fig. 4c), and in fact, MERRA and MERRA-2 midlatitude profiles of AoA agree best with the observations.

You keep referring to MERRA-2 as an "outlier", which carries negative connotations, but in fact being an outlier in this comparison is a good thing if one considers the data to be the "truth".

(9, 14) "MERRA-2 starts with much older values": This behavior does appear to be anomalous. Any idea what might be causing it?

(9, 18) "The Pinatubo eruption does not appear to have any impact of the simulated AoA at 50 hPa": Insofar as one might expect that the largest impact of Pinatubo would be in the Tropics, it might be worthwhile to examine the AoA time series averaged over, say, 30N-30S.

(10, 8) "observational trend is not significant": One would not expect any trend calculated from the smoothed, model time series shown on the right pane of Figure 8 to be significant either. By the way, you keep referring to "trends" in connection with the model results, but you have not calculated any trends. Note also that Garcia et al. (2011) have argued that, even using model output for an ideal AoA tracer, trends over periods as long as 30 years are often not significant when the ideal tracer is sampled like the available observations of stratospheric tracers. Furthermore, trends derived from observation are also confounded by the fact that no real atmospheric tracer has a constant, linear growth rate.

(11,16) "in the polar regions and midlatitudes": Why limit this to extratropical behavior only? It would be interesting to show the seasonal amplitude in the Tropics as well, say a $\pm30°$ average.

(11, 24) "MERRA and MERRA-2 . . . different amplitudes depending on the period used for the analysis": Does this have to do with the development (1989-2001) and stabilization/decline (2002-2015) of the Antarctic ozone hole? To explore this issue, one would have to examine the actual seasonal climatology at high SH latitudes, not just the annual amplitude.

(12, 9) "could not be expected from inspection of the native dynamic variables": I do not understand what you are trying to say here. Please elaborate. And note also that the discrepancies you mention ("up to 50% dependencies on the considered time period") are not even illustrated, so it is very difficult to even guess what the intent of your statement is.

(12, 15) "unexpectedly increasing": Given the very short period covered by the SF6 observations, it is not clear that one should "expect" any particular sign for the trends. Determination of AoA trends from observations of stratospheric tracers is fraught with many uncertainties; even in models where an ideal, linearly increasing artificial tracer is used, one has to rely on zonal-mean results over long periods to obtain trends that are clearly statistically significant. Arguably, examination of AoA trends determined from observations of stratospheric tracers is not the best tool for documenting changes in the BD circulation. See Garcia et al. (2011)

(13, 1) "unexpected growth": Again, there are no clear expectations about trends for short periods.

(13, 23) "the reversal is found for all five reanalyses": This is true, but the reversals are in the opposite sense in ERAi and CFS vs. JRA-55, MERRA and MERRA2, so it is hard to know what to make of this.

(13, 25) "unexpectedly large disagreements": I am not sure why you think the disagreements are "unexpected". While presumably all reanalyses use more or less the same observational data, the manner in which the data are assimilated and the physical parameterizations included in the different models (in particular, those for convection and mesoscale gravity waves) are different. Note that at (14, 8) you suggest that "the disagreements found here may lie in the differences between the underlying models"; I agree that this is the most plausible working hypothesis.

Grammar, typos, etc.

(2, 21) "monotonously" → monotonically

(3, 10) "technical difficulties which" → technical difficulties that

(6, 27) "very similar than" → very similar to

(9, 33) "twice smaller" → twice as small

(11, 30) "between reanalyses" → among reanalyses → among reanalyses

References cited:

Garcia, R. R., W. J. Randel and D. E. Kinnison, 2011: On the determination of age of air trends from atmospheric trace species, J. Atmos. Sci., 68, 139-154, DOI: 10.1175/2010JAS3527.1.

Waugh, D. W. and T, M. Hall, 2002: Age of stratospheric air: Theory, observations and models, Rev. Geophys., 40, doi:10.1029/2000RG000101.

---

## Author Comment (AC1) · 7 Sep 2018

**Response to Reviewer #1 for discussion paper**

**Comparison of mean age of air in five reanalyses using the BASCOE transport model**

**Chabrillat et al., ACPD, 2018**

We thank the reviewer for his/her positive and useful comments. In our replies below the bold type is used to highlight text in the revised manuscript.

- *It is stated that the model input is wind and surface pressure. Is the latter used to convert from model levels to pressure levels? Perhaps this could be said directly at some point.*

Yes: the output AoA datasets are indeed interpolated from model levels to constant pressure levels using the instantaneous and two-dimensional input surface pressures, i.e. prior to any averaging in the longitudinal or time dimension. This is now stated explicitly at the end of section 2.1.

- *Is the sentence in L15-16 (P1) of the abstract needed? It seems redundant.*

This sentence has been removed from the abstract.

- *L4 P2: "considerably depending on the considered period": consider changing one of the two to avoid repetition.*

"Considerably" has been replaced by "substantially"

- *L30 P3: explain how these...*

Typo corrected

- *L32 P3: each tailored for a different reanalysis dataset.*

This sentence was outdated because the version published in ACPD compared with only one other transport model (TOMCAT) which is tailored for ERA-I. We have updated the sentence accordingly.

- *L14-15 P4: It could be mentioned why it is chosen not to use the vertical wind component directly.*

The revised manuscript states:
> **These models are usually set on a different grid than their input reanalysis dataset. Since this prevents the direct usage of the vertical wind component in the reanalysis, they rely on mass continuity to derive the vertical mass fluxes corresponding to their own grid.**

- *L31 P4: "idealized tracer which increases linearly at the surface": throughout the surface or just in the tropics?*

The choice of the surface as source region introduced confusing inconsistencies in the discussion paper (see the general comment by the second referee and also the next comment here). Hence we decided to re-run our calculations and re-plot all figures (except for figure 8, see below) using as source region the tropical tropopause region (defined as the 100 hPa isobar between latitudes 10°S and 10°N), and computing the AoA at each gridpoint as the time elapsed since the mixing ratio of the ideal tracer reached the same value in that source region. The figures did not change significantly from the discussion paper, indicating that this is a methodological issue which does not have any impact on our findings. The last paragraph of section 2.1 has been re-written to fully explain the updated procedure for computing AoA.

For figure 8 (and figure 8 only) we have kept the original calculations where the tracer was set to increase linearly *throughout* the surface, because this figure includes a comparison with observational values of AoA which used the surface as reference. We have moved to the disussion of figure 8 the description of this *surface* boundary condition and its propagation through the troposphere, because it is now irrelevant for all other figures. This description now states explicitly that it uses

> **... a synthetic tracer which is set to a global constant increasing linearly with time at the surface.**

- *L31-32 P6: "the AoA at the equatorial tropopause has been subtracted from the fields...": did you use the climatological or time-dependent tropopause altitude?*

See previous question: the revised manuscript shows AoA computes AoA directly from the tropical tropopause region and has dropped all *a posteriori* corrections by subtraction of AoA values at the equatorial tropopause. We have removed the sentences describing this procedure from the revised manuscript.

- *L9 P8: GCCM: this has not been introduced before, do you mean CCM?*

Yes. We have replaced all occurrences of "GCCM" by "CCM".

- *L14-15 P8: "different latitude gradients between the tropics and ..."*

Done.

- L13 P9: remove "globally", it is not global but midlatitude average.

Done.

- *L30 and 31 P9: "different with" should be "different from"*

Done.

- *L11 P10: "not significant": in ERA-Interim or in observations?*

Replaced "sign of observational trend not significant" by clearer

**...sign of trends not significant in the observations.**

- *L12 P10: "ERA-I does not show any overall trend after 2000...". Why do you point out these trends after 2000 in ERA-I specifically? Same thing on P16 L22-23.*

This focus on ERA-I is due to the exclusive use of ERA-I in previous studies modeling the latitudinal structure of AoA for the post-2000 period (see first paragraph of section 4.3, P12 of ACPD manuscript). But this context had not been introduced yet for the discussion of figure 8 (i.e. L12 P10 and P16 L22-23). Since this question is specifically investigated through figures 11 and 12, we have simply removed the premature sentence from the discussion of figure 8.

- *L23 P12: standard error for which confidence level?*

This important information has been added in section 4.1 which describes our methodology for multi-linear regressions:
> **The uncertainties arising from the fit are calculated for the 95% confidence interval and corrected for auto-correlation in the residuals (Eqs. 3,4 and 6 in Santer et al., 2000).**

and in the section 4.3 (discussing linear trends):
> **It is expressed in years per decade (yr dec$^{-1}$) and is deemed significant at a given grid point if its absolute value is larger than its uncertainty (as defined in section 4.1).**

- *L1 P13: remove "unexpected".*

Done.

- *L2 P13: remove "much"*

Done.

- *L7 P13: "Diallo et al. (2012)..." Ploeger et al. (2015a) show AoA trends for the total and later periods.*

Thanks for pointing this out. In the revised manuscript we now also compare our results with those by Ploeger et al. (2015a), both for the discussion of the latter period (figure 11):
> **Our results also agree well with those obtained by a diabatic model driven by ERA-I over the same period (Ploeger et al., 2015a).**

and for the discussion of the total period (first paragraph discussing figure 12):
> **Our ERA-I results for the overall period partly contradict those obtained by diabatic models which use not only the wind fields from ERA-I but also its heating rates (Diallo et al., 2012; Ploeger et al., 2015a). Looking at slightly shorter periods of two decades (1989–2010 for the former and 1990–2013 for the latter), these papers reported negative AoA trends for both hemispheres below 28km altitude.**

- *L10 P13: "using only wind fields": do you mean not using heating rates? Perhaps this should be explicit.*

Indeed we meant that our results did not use the heating rates. During our revision we found that the whole sentence was confusing and removed it from the manuscript. The additional use of ERA-I heating rates by the diabatic models (Diallo et al., 2012; Ploeger et al., 2015a) is now explicitly

stated as soon as they are cited (see previous comment).

- *L26-27 P14: "While this may be a coincidence...": but having more wave drag would imply a faster BDC, so I do not see the point of this sentence.*

Agreed. This sentence has been removed from the revised manuscript.

- *L34 P14: "Miyazaki et al. (2016) Fig. 11": The trends in annual mean tropical upwelling for these reanalyses are shown in Abalos et al. (2015) Fig. 11.*

The following sentence has been added to the discussion:
> **Similar disagreements have also been reported between the trends of the annual mean tropical upwelling in three reanalyses over the period 1979–2012, with vertical residual velocities ($\overline{w}$*) increasing in MERRA and JRA-55 and decreasing in ERA-I (Abalos et al., 2015, Fig. 11).**

- *L18 P15: Another difference with CLaMS is that it includes a mixing parameterization.*

The revised version states also this difference, citing Konopka et al.
(JGR, doi:10.1029/2003JD003792, 2004).

- *Figure 7 caption: "No impact..." This sentence does not belong to the caption but to the main text.*

This sentence has been removed from the caption of Fig.~7.

**Added references**

Konopka, P., Steinhorst, H.-M., Grooß, J.-U., Günther, G., Müller, R., Elkins, J. W., Jost, H.-J., Richard, E., Schmidt, U., Toon, G., and McKenna, D. S.: Mixing and ozone loss in the 1999–2000 Arctic vortex: Simulations with the three-dimensional Chemical Lagrangian Model of the Stratosphere (CLaMS), Journal of Geophysical Research: Atmospheres, 109, doi:10.1029/2003JD003792, 2004.

---

## Author Comment (AC2) · 8 Sep 2018

**Response to Reviewer #2 for discussion paper**

**Comparison of mean age of air in five reanalyses using the BASCOE transport model**

**Chabrillat et al., ACPD, 2018**

We thank the reviewer for his/her insightful comments. It appears that the version of the manuscript which was reviewed by this referee is the version first submitted to ACPD (on 4 April 2018) rather than the version finally published in ACPD (on 7 May 2018). Fortunately all comments apply equally to both versions. In our replies below the bold type is used to highlight text in the revised manuscript.

**Replies to general comments**

- *The principal conclusion is that the simulations of AoA obtained when BASCOE is constrained by different reanalyzed datasets differ substantially from one another. This is not at all unexpected given the differences among the reanalysis models.*

The reanalysis systems are based on different models but they assimilate very similar satellite datasets. Many users of reanalyses are neither modellers of stratospheric dynamics nor aware of the lack of observational information to constrain the BDC in the reanalyses. From feedback obtained at the 5th International Conference on Reanalysis (ICR5, Rome, November 2017), such users do not expect to see a spread between the reanalyses which is as large as between unconstrained GCCMs (Fig. 4). On Fig. 8 they easily understand that the uncertainties in the observational timeseries are large (due to sparse and irregular sampling) but they do not expect to see that the spread between the reanalyses is as large as these observational uncertainties.

A third highlight of this paper is the intercomparison of AoA trends between the reanalyses. Several reanalysis intercomparisons of diagnostics related to stratospheric dynamics have already been published and showed significant differences with respect to their trends (e.g. Abalos et al., 2015; Miyazaki et al., 2016). Yet for the AoA diagnostic, most recent studies rely on ERA-I with much interest in the latitudinal structure of its trends. We found that over the post-2002 period ERA-I is the *only* reanalysis to deliver opposite trends of AoA in the two hemispheres (Fig. 12, middle column). This is also an unexpected result.

- *The paper is well organized and clearly written, with some exceptions, the main one being that the procedure for computing AoA is not well explained. In particular, it is not clear whether AoA is calculated with respect to a reference level at the tropical tropopause or in the troposphere, and this introduces some ambiguity in the interpretation of the results.*

We agree with the referee that the handling of the reference level was problematic in the submitted manuscript. All our calculations used the surface both as the source region and to compute the time lag defining the mean Age of Air. But in order to better highlight the different transit times from the equatorial tropopause, Fig.1 and 3 were corrected a posteriori by subtraction of the time-averaged AoA at 100hPa, 10°S-10°N. All other figures used the surface as reference, hence including the transit time from the surface to the tropical tropopause. This distinction was not clearly made and led to inconsistent figures, as shown by several specific comments made by both reviewers.

Hence we decided to re-run our calculations and re-plot all figures (except for figure 8, see below) using as source region the tropical tropopause region (still defined as the 100 hPa isobar between latitudes 10°S and 10°N), and computing the AoA at each grid point as the time elapsed since the mixing ratio of the ideal tracer reached the same value in that source region. The last paragraph of section 2.1 has been re-written to fully explain the updated procedure for computing AoA.

The figures did not change much from the discussion paper, indicating that this is a methodological issue which does not have a large impact on our findings. Besides figures 3 and 9 which are discussed below for specific comments, there is one other case where the figure changed sufficiently to warrant a minor update in the text: on figure 12 the positive AoA trends for ERA-I in 2002-2015 (top row, middle column) have become significant at all latitudes (in the discussion paper they were significant only in the polar latitudes). On figure 12 the signs and patterns of AoA trends did not change for any other reanalysis or period but the range of these trends increased by up to 50% (see min/max values above the plots); this led us to extend the scale of the color bar, from [-0.6,0.6] to [-0.9,0.9].

For figure 8 (and figure 8 only) we have kept the original calculations where the tracer was set to increase linearly throughout the surface, because this figure includes a comparison with observational values of AoA which used the surface as reference. We have moved to the discussion of figure 8 the description of this surface boundary condition and its propagation through the troposphere, because it is now irrelevant for all other figures. We have also added in this figure a plot showing tropical AoA computed both from the surface and from the tropical tropopause to show that the difference does not vary significantly with the simulated year (see next comment).

**Replies to specific comments**

- *(4, 20) "There is no other representation of convection": It is not clear that, in the Tropics, where deep convection can reach the 14-15 km level, this artificial diffusion can simulate vertical transport realistically. But perhaps this does not matter for assessments of AoA in the stratosphere if the base point for AoA calculations is taken to be at or near the tropical tropopause? Please comment (especially since it is not clear how the reference level for computing AoA is chosen). See also comment at (7, 24).*

As explained above, nearly all figures now use (100hPa, 10°S-10°N) as reference region hence the absence of deep convection in the CTM is not an issue for the updated figures. Since for figure 8 we keep using the surface as reference, we added tropical timeseries in the tropics which show both the surface-based evaluation (solid lines) and the tropopause-based evaluation (dashed lines):

[Figure]

**Middle panel of revised Fig. 8**. Mean AoA in the mid-stratosphere (5-30hPa) for the tropical latitudes (30°S–30°N). Solid lines show AoA using the surface as reference, dashed lines show AoA using the tropical tropopause as reference (i.e. as in all other figures of the revised manuscript).

This comparison between the two evaluations also allows a discussion on the impact of the omission of deep convection in the model. The discussion of Fig. 8 in the revised manuscript includes the following paragraph:

**These differences between the two calculations represent the transit times from the surface to the tropopause, are nearly independent of the simulated year and range between 3 months (with ERA-I or JRA-55) and 6 months (with MERRA). These values are close to the longest transit times reported in a recent intercomparison of global models (Krol et al., 2018), indicating a rather slow transport from the surface to the tropical tropopause which we attribute to the omission of deep convective transport in our model. Hence the model results in Fig. 8 may be slightly overestimated but these biases have no significant inter-annual variations and do not hinder the intercomparison between reanalyses.**

- *(6, 5) "at the wavelength number 47": "at wavenumber 47" might be better.*

Text corrected.

- *(6, 8) "Figure 1 compares the results": I do not believe you have stated how AoA is calculated. [...]*

Section 2.1 now describes explicitly the revised procedure to calculate AoA from the tropical tropopause region:

**The age of air is defined as the spectrum of transit times from a source region to a given location, with the tropical tropopause usually defining the source region for studies of the stratosphere. In the case of ideal tracers which increase linearly in the source region and have no photochemical production or losses, the mean of this spectrum (denoted here AoA) is simply the time elapsed since the mixing ratio of this ideal tracer reached the same value in the source region (see e.g. Waugh and Hall, 2002). We follow here this classical approach, using for most simulations the 100 hPa isobar between latitudes 10°S and 10°N as source region.**

Section 3.2 describes explicitly the original procedure which has been kept only for figure 8:

**For consistency the modeled AoA in this figure are evaluated as the time elapsed since the mixing ratio of an ideal tracer reached the same value at the surface, using as boundary condition a global constant increasing linearly with time at the surface.**

- *(7, 18) Figure 2: This figure, as well as Figure 3, would benefit from a color bar to indicate the values of the AoA isolines not explicitly labeled.*

A color bar has been added to figures 2 and 3.

- *(7, 24) Figure 3: I am confused by this comparison [...] So, where is the reference point in these simulations? If it is at the surface, then AoA will reflect the effects of transport not just in the stratosphere, but also in the troposphere, including the artificial diffusive transport between the surface and the middle troposphere. Unless I am misunderstanding what you have done here, it seems to me that, if AoA is intended to highlight transport in the stratosphere (e.g., Waugh and Hall, 2002, Sec. 3.1) then the choice of a base point in the troposphere confuses the issue, especially given the use of artificial diffusive transport in the lower troposphere.*

The reviewer was rightly confused and his interpretation is correct. We have followed this advice, choosing the tropical tropopause as reference point in the revised manuscript (see above). The relative differences between ERA-I and the four other reanalyses vanish at the reference point and the difference is not plotted at grid points where ERA-I AoA is smaller than 5 days.

- *(8, 5) "The intercomparison at 50 hPa": You should state explicitly in the text that in these comparisons AoA is "normalized" to be zero at the tropical tropopause (this is only stated in the caption of Figure 4). Otherwise, the reader will wonder, as I did, why the AoA shown in Figures 2-3 are different from the AoA in Figure 4. By the way, a problem with the "normalization" of AoA to zero at 100 hPa is that it gives the impression that AoA above that level is determined only by the stratospheric circulation, when in fact the AoA also contains the effect of transport in the troposphere.*

thanks to the direct calculation of AoA using the tropical tropopause as reference point, no "normalization" is performed any more for the figures 1 and 4 of the revised manuscript.

- *(8, 12) "overall, the spread . . . is larger than the 1-sigma. . .": One wonders how this result would change if AoA were computed with respect to a reference point at 100 hPa.*

In the revised manuscript the AoA are computed with respect to a reference point at 100 hPa. The differences in Figure 4 between the submitted and revised manuscripts are nearly indistinguishable. Hence the spread between the five simulations at 50 hPa is still larger than the 1-$\sigma$ observational uncertainties in the tropics, and still nearly as large in the extratropics. We have not modified this sentence in the revised manuscript.

- *(8, 26) "The spread between the four reanalyses reaches a maximum of 0.2 years at 30 hPa": Are you referring here to the gradient comparison, Figure 4d? How is this "gradient" calculated? The figure legend refers to "MLNH-Tropics" and shows values in months, not per unit distance, so this is really a difference between the Tropics andmidlatitudes o f the NH. How are Tropics and NH midlatitudes defined?*

The words "(mean age) gradient profiles" or "latitudinal gradients (of mean age)" were meant with the same meaning as Neu et al. (2010) and Chipperfield et al. (2014) i.e. as the difference between AoA in NH midlatitudes and AoA in the Tropics. The vertical profiles on figure 4d simply show the differences between the corresponding profiles on figures 4c and 4b which are mean values for latitude bands 35°N-45°N and 10°S-10°N respectively (as stated in the figure of caption 4).

In the revised manuscript we have added the definition of the latitude bands in the discussion of figures 4b and 4c and we have added the following sentences in the discussion of figure 4d:
> **These "latitudinal gradients of AoA" were used in several CCM intercomparisons (Neu et al., 2010; Chipperfield et al., 2014). Figure 4d shows this diagnostic for the five reanalyses, i.e. the differences between the AoA profiles on Fig. 4c and Fig. 4b.**

We have replaced the words "latitudinal gradients" by "AoA differences" in the remainder of this discussion and in the caption of the figure.

- *(8, 30) "MERRA-2 yields an outlying vertical profile of AoA at northern midlatitudes": True with respect to the other reanalyses except for MERRA (Fig. 4c), and in fact, MERRA and MERRA-2 midlatitude profiles of AoA agree best with the observations. You keep referring to MERRA-2 as an "outlier", which carries negative connotations, but in fact being an outlier in this comparison is a good thing if one considers the data to be the "truth"..*

Thanks for pointing this out. We have corrected the discussion according to your comment:
**MERRA and MERRA-2 yield larger AoA at northern midlatitudes than the three other reanalyses. In the case of MERRA-2 this results in a profile of AoA differences which are significantly larger than the profiles obtained with the four other reanalyses but agrees much better with the profile derived from the observations. Hence MERRA-2 apparently underestimates the tropical upwelling in the lowermost stratosphere (100-60~hPa), agrees better with the observations at 50~hPa than any other simulation, and joins the results of the four other reanalyses at higher levels.**

- *(9, 14) "MERRA-2 starts with much older values": This behavior does appear to be anomalous. Any idea what might be causing it?*

This issue is discussed in detail in sectin 5 (see paragraph starting with "MERRA-2"). We have inserted the following sentence in section 3.2:
**The possible causes for this apparently anomalous behavior of MERRA-2 are discussed in section 5.**

- *(9, 18) "The Pinatubo eruption does not appear to have any impact of the simulated AoA at 50 hPa": Insofar as one might expect that the largest impact of Pinatubo would be in the Tropics, it might be worthwhile to examine the AoA time series averaged over, say, 30N-30S.*

This was done and no impact was found for the tropical latitude band, as shown by the corresponding plot:

[Figure]

Note that any impact in the 30°S-30°N latitude band would have been seen in figure 7 which shows the 72°S-72°N band. The revised manuscript mentions the absence of volcanic impact in the tropical latitude band as well.

- *(10, 8) "observational trend is not significant": One would not expect any trend calculated from the smoothed, model time series shown on the right pane of Figure 8 to be significant either. By the way, you keep referring to "trends" in connection with the model results, but you have not calculated any trends...*

Not yet. As outlined in the introduction, modelled trends are evaluated extensively in section 4. The sentence now refers to this later finding:

> **ERA-I delivers a weakly positive trend over the period 1989-2015 and we will assess in section 4.3 that this trend in the model results is significant.**

And this finding has been added in the discussion of figure 12 (section 4.3):

> **The same plot** (*i.e. Fig. 12, top right)* **also shows that the positive trend which had been inferred visually for the northern mid-latitudes of the middle stratosphere (Fig. 8, left) is significant.**

- *...Note also that Garcia et al. (2011) have argued that, even using model output for an ideal AoA tracer, trends over periods as long as 30 years are often not significant when the ideal tracer is sampled like the available observations of stratospheric tracers....*

Great caution should indeed be exercised in comparisons of trends between model output and observational datasets which are sparsely and irregularly sampled. This is what we meant in the original manuscript with "..., but Engel et al. (2009, 2017) showed that the sign of this observational trend is not significant". Referring to Garcia et al. (2011) allows us to reinforce and clarify this call to caution. The revised manuscript states:

> **While the overall trend simulated with ERA-I is apparently in agreement with the balloon observations, this comparison should be considered with great caution because the sign of the AoA trend is not significant in the observations (Engel et al., 2009, 2017) and modelled trends over periods as long as 30 years are often not significant when the ideal tracer is sampled like the available observations of stratospheric tracers (Garcia et al., 2011).**

- *... Furthermore, trends derived from observation are also confounded by the fact that no real atmospheric tracer has a constant, linear growth rate.*

The non-linearity of the growth rate of $CO_2$ and $SF_6$ has of course been taken into account by Engel et al. (2009, 2017) for their derivation of AoA and also for their error analysis. The procedure is described in the supplementary information of Engel et al. (2009) and in the first paragraph of section 4 in Engel et al. (2017).

- *(11,16) "in the polar regions and midlatitudes": Why limit this to extratropical behavior only? It would be interesting to show the seasonal amplitude in the Tropics as well, say a ±30° average.*

Thanks for the suggestion. The seasonal amplitudes with ERA-I and MERRA-2 have a different vertical structure in the Tropics. We have added this plot to figure 9 in the revised manuscript.

- *(11, 24) "MERRA and MERRA-2 . . . different amplitudes depending on the period used or the analysis": Does this have to do with the development (1989-2001) and stabilization/decline (2002-2015) of the Antarctic ozone hole? To explore this issue, one would have to examine the actual seasonal climatology at high SH latitudes, not just the annual amplitude.*

The left plot below is extracted from fig. 9 in the ACPD manuscript while the right plot shows results with the new AoA calculation (i.e. using the tropical tropopause as reference):

[Figure]

This shows that the new AoA calculation delivers amplitudes of AoA seasonal variations which are much closer for periods 1989-2001 and 2002-2015. This is the case also for the other latitude bands, so for the revised manuscript we have removed from figure 9 the results for the early period 1989-2001 and we have dropped the corresponding part of the discussion.

- *(12, 9) "could not be expected from inspection of the native dynamic variables": I do not understand what you are trying to say here. Please elaborate. And note also that the discrepancies you mention ("up to 50% dependencies on the considered time period") are not even illustrated, so it is very difficult to even guess what the intent of your statement is.*

The dependencies of seasonal amplitudes on the considered time period reached 50% in fig. 9a and 9c of the ACPD manuscript but, as explained in the previous comment, they have disappeared from the revised manuscript thanks to the corrected AoA calculation. We removed the whole paragraph (i.e. last paragraph of section 4.2) from the revised manuscript because it is redundant with the second paragraph of section 5.

- *(12, 15) "unexpectedly increasing": Given the very short period covered by the $SF_6$ observations, it is not clear that one should "expect" any particular sign for the trends. Determination of AoA trends from observations of stratospheric tracers is fraught with many uncertainties; even in models where an ideal, linearly increasing artificial tracer is used, one has to rely on zonal-mean results over long periods to obtain trends that are clearly statistically significant. Arguably, examination of AoA trends determined from observations of stratospheric tracers is not the best tool for documenting changes in the BD circulation. See Garcia et al. (2011).*

We acknowledge that it is important to provide proper context about this topic. Section 4.3 now starts with a new paragraph explaining some caveats of interpreting AoA trends as changes in BDC. The next paragraph has been expanded and corrected to justify the interest of the section:

**It is delicate to infer changes in the BDC from the examination of AoA trends which are derived from observations of stratospheric tracers over periods shorter than several decades. Even in models where an ideal, linearly increasing artificial tracer is used, one has to rely on zonal-mean results over long periods to obtain trends that are clearly statistically significant (Garcia et al., 2011). The statement that is often made that climate models simulate a decreasing AoA throughout the stratosphere only applies over long time periods and is not necessarily the case for the past 25 years, when most tracer measurements were taken (Garfinkel et al., 2017). For example, the analysis of a 1700 year simulation showed that it takes around 30 years for a modeled BDC trend to emerge from the noise of natural climate variability (assuming a 2%/decade trend in the BDC; Hardiman et al., 2017).**

**While linear trends of AoA over shorter periods may represent transient changes due to climate variability, changes over such intermediate timescales (i.e. intermediate between the QBO and the multidecadal scales) are still relevant to the study of stratospheric dynamics.** Current research on AoA trends has largely focused on a dipole-like latitudinal structure for the period 2002-2012, which was first derived from satellite observation of $SF_6$ by the MIPAS instrument (Stiller et al., 2012). This structure of trends shows AoA decreasing in the Southern Hemisphere but  increasing in the Northern Hemisphere **which was used to explain a recent increase of stratospheric HCl in the Northern Hemisphere (Mahieu et al., 2014) and interpreted as the** consequence of a southward shift of the subtropical transport barriers (Stiller et al., 2017) .

- *(13, 1) "unexpected growth": Again, there are no clear expectations about trends for short periods.*

We have removed the word "unexpected" from this sentence.

- *(13, 23) "the reversal is found for all five reanalyses": This is true, but the reversals are in the opposite sense in ERAi and CFS vs. JRA-55, MERRA and MERRA2, so it is hard to know what to make of this.*

Yes but we still want to highlight this striking feature in case some reader does find what to make of this. We have changed this sentence to:

This reversal is found for all five reanalyses and in all regions of the stratosphere **but it is difficult to interpret because it goes in opposite directions in ERA-I and CFSR versus JRA-55, MERRA and MERRA-2.**

- *(13, 25) "unexpectedly large disagreements": I am not sure why you think the disagreements are "unexpected". While presumably all reanalyses use more or less the same observational data, the manner in which the data are assimilated and the physical parameterizations included in the different models (in particular, those for convection and mesoscale gravity waves) are different. Note that at (14, 8) you suggest that "the disagreements found here may lie in the differences between the underlying models"; I agree that this is the most plausible working hypothesis.*

See above the reply to the first general comment. The word "unexpectedly" had already been removed before publication in ACPD.

**Grammar, typos, etc.**

All errors have been corrected.

**Added references**

Garcia, R. R., Randel, W. J., and Kinnison, D. E.: On the Determination of Age of Air Trends from Atmospheric Trace Species, J. Atmos. Sci., 68, 139–154, doi:10.1175/2010JAS3527.1, 2011.

Garfinkel, C. I., Aquila, V., Waugh, D. W., and Oman, L. D.: Time-varying changes in the simulated structure of the Brewer–Dobson Circulation, Atmospheric Chemistry and Physics, 17, 1313–1327, doi:10.5194/acp-17-1313-2017, 2017.

Hardiman, S. C., Lin, P., Scaife, A. A., Dunstone, N. J., and Ren, H.-L.: The influence of dynamical variability on the observed Brewer-Dobson Circulation trend, Geophys. Res. Lett., doi:10.1002/2017GL072706, 2017.

Krol, M., de Bruine, M., Killaars, L., Ouwersloot, H., Pozzer, A., Yin, Y., Chevallier, F., Bousquet, P., Patra, P., Belikov, D., Maksyutov, S., Dhomse, S., Feng, W., and Chipperfield, M. P.: Age of air as a diagnostic for transport timescales in global models, Geoscientific Model Development, 11, 3109–3130, doi:10.5194/gmd-11-3109-2018, 2018.

---

## Author Response (AR2)

Brussels, 13 September 2018

Dear William,

I have now prepared the replies and changed the text according to the reviewer's comments.

During the review process we found that it is more consistent for a model intercomparison to calculate the mean age of air using as reference the tropical tropopause. The figures did not change much from the discussion paper, indicating that this is a methodological issue which does not have a large impact on our findings. In the reply to the second reviewer I explain all the details about this change and its impacts on the manuscript.

Sincerely,

Simon Chabrillat (on behalf of all co-authors).

*Response to Reviewer #1 for discussion paper*

**Comparison of mean age of air in five reanalyses using the BASCOE transport model**

*Chabrillat et al., ACPD, 2018*

We thank the reviewer for his/her positive and useful comments. In our replies below the bold type is used to highlight text in the revised manuscript.

- *It is stated that the model input is wind and surface pressure. Is the latter used to convert from model levels to pressure levels? Perhaps this could be said directly at some point.*

Yes: the output AoA datasets are indeed interpolated from model levels to constant pressure levels using the instantaneous and two-dimensional input surface pressures, i.e. prior to any averaging in the longitudinal or time dimension. This is now stated explicitly at the end of section 2.1.

- *Is the sentence in L15-16 (P1) of the abstract needed? It seems redundant.*

This sentence has been removed from the abstract.

- *L4 P2: "considerably depending on the considered period": consider changing one of the two to avoid repetition.*

"Considerably" has been replaced by "substantially"

- *L30 P3: explain how these...*

Typo corrected

- *L32 P3: each tailored for a different reanalysis dataset.*

This sentence was outdated because the version published in ACPD compared with only one other transport model (TOMCAT) which is tailored for ERA-I. We have updated the sentence accordingly.

- *L14-15 P4: It could be mentioned why it is chosen not to use the vertical wind component directly.*

The revised manuscript states:
**These models are usually set on a different grid than their input reanalysis dataset. Since this prevents the direct usage of the vertical wind component in the reanalysis, they rely on mass continuity to derive the vertical mass fluxes corresponding to their own grid.**

- *L31 P4: "idealized tracer which increases linearly at the surface": throughout the surface or just in the tropics?*

The choice of the surface as source region introduced confusing inconsistencies in the discussion paper (see the general comment by the second referee and also the next comment here). Hence we decided to re-run our calculations and re-plot all figures (except for figure 8, see below) using as source region the tropical tropopause region (defined as the 100 hPa isobar between latitudes 10°S and 10°N), and computing the AoA at each gridpoint as the time elapsed since the mixing ratio of the ideal tracer reached the same value in that source region. The figures did not change significantly from the discussion paper, indicating that this is a methodological issue which does not have any impact on our findings. The last paragraph of section 2.1 has been re-written to fully explain the updated procedure for computing AoA.

For figure 8 (and figure 8 only) we have kept the original calculations where the tracer was set to increase linearly *throughout* the surface, because this figure includes a comparison with observational values of AoA which used the surface as reference. We have moved to the disussion of figure 8 the description of this *surface* boundary condition and its propagation through the troposphere, because it is now irrelevant for all other figures. This description now states explicitly that it uses

> **... a synthetic tracer which is set to a global constant increasing linearly with time at the surface.**

- *L31-32 P6: "the AoA at the equatorial tropopause has been subtracted from the fields...": did you use the climatological or time-dependent tropopause altitude?*

See previous question: the revised manuscript shows AoA computes AoA directly from the tropical tropopause region and has dropped all *a posteriori* corrections by subtraction of AoA values at the equatorial tropopause. We have removed the sentences describing this procedure from the revised manuscript.

- *L9 P8: GCCM: this has not been introduced before, do you mean CCM?*

Yes. We have replaced all occurrences of "GCCM" by "CCM".

- *L14-15 P8: "different latitude gradients between the tropics and ..."*

Done.

- L13 P9: remove "globally", it is not global but midlatitude average.

Done.

- *L30 and 31 P9: "different with" should be "different from"*

Done.

- *L11 P10: "not significant": in ERA-Interim or in observations?*

Replaced "sign of observational trend not significant" by clearer

**...sign of trends not significant in the observations.**

- *L12 P10: "ERA-I does not show any overall trend after 2000...". Why do you point out these trends after 2000 in ERA-I specifically? Same thing on P16 L22-23.*

This focus on ERA-I is due to the exclusive use of ERA-I in previous studies modeling the latitudinal structure of AoA for the post-2000 period (see first paragraph of section 4.3, P12 of ACPD manuscript). But this context had not been introduced yet for the discussion of figure 8 (i.e. L12 P10 and P16 L22-23). Since this question is specifically investigated through figures 11 and 12, we have simply removed the premature sentence from the discussion of figure 8.

- *L23 P12: standard error for which confidence level?*

This important information has been added in section 4.1 which describes our methodology for multi-linear regressions:

> **The uncertainties arising from the fit are calculated for the 95% confidence interval and corrected for auto-correlation in the residuals (Eqs. 3,4 and 6 in Santer et al., 2000).**

and in the section 4.3 (discussing linear trends):

> **It is expressed in years per decade (yr dec$^{-1}$) and is deemed significant at a given grid point if its absolute value is larger than its uncertainty (as defined in section 4.1).**

- *L1 P13: remove "unexpected".*

Done.

- *L2 P13: remove "much"*

Done.

- *L7 P13: "Diallo et al. (2012)..." Ploeger et al. (2015a) show AoA trends for the total and later periods.*

Thanks for pointing this out. In the revised manuscript we now also compare our results with those by Ploeger et al. (2015a), both for the discussion of the latter period (figure 11):

> **Our results also agree well with those obtained by a diabatic model driven by ERA-I over the same period (Ploeger et al., 2015a).**

and for the discussion of the total period (first paragraph discussing figure 12):

> **Our ERA-I results for the overall period partly contradict those obtained by diabatic models which use not only the wind fields from ERA-I but also its heating rates (Diallo et al., 2012; Ploeger et al., 2015a). Looking at slightly shorter periods of two decades (1989–2010 for the former and 1990–2013 for the latter), these papers reported negative AoA trends for both hemispheres below 28km altitude.**

- *L10 P13: "using only wind fields": do you mean not using heating rates? Perhaps this should be explicit.*

Indeed we meant that our results did not use the heating rates. During our revision we found that the whole sentence was confusing and removed it from the manuscript. The additional use of ERA-I heating rates by the diabatic models (Diallo et al., 2012; Ploeger et al., 2015a) is now explicitly

stated as soon as they are cited (see previous comment).

- *L26-27 P14: "While this may be a coincidence...": but having more wave drag would imply a faster BDC, so I do not see the point of this sentence.*

Agreed. This sentence has been removed from the revised manuscript.

- *L34 P14: "Miyazaki et al. (2016) Fig. 11": The trends in annual mean tropical upwelling for these reanalyses are shown in Abalos et al. (2015) Fig. 11.*

The following sentence has been added to the discussion:
**Similar disagreements have also been reported between the trends of the annual mean tropical upwelling in three reanalyses over the period 1979–2012, with vertical residual velocities ($\overline{w}*$) increasing in MERRA and JRA-55 and decreasing in ERA-I (Abalos et al., 2015, Fig. 11).**

- *L18 P15: Another difference with CLaMS is that it includes a mixing parameterization.*

The revised version states also this difference, citing Konopka et al.
(JGR, doi:10.1029/2003JD003792, 2004).

- *Figure 7 caption: "No impact..." This sentence does not belong to the caption but to the main text.*

This sentence has been removed from the caption of Fig.~7.

**Added references**

Konopka, P., Steinhorst, H.-M., Grooß, J.-U., Günther, G., Müller, R., Elkins, J. W., Jost, H.-J., Richard, E., Schmidt, U., Toon, G., and McKenna, D. S.: Mixing and ozone loss in the 1999–2000 Arctic vortex: Simulations with the three-dimensional Chemical Lagrangian Model of the Stratosphere (CLaMS), Journal of Geophysical Research: Atmospheres, 109, doi:10.1029/2003JD003792, 2004.

*Response to Reviewer #2 for discussion paper*

**Comparison of mean age of air in five reanalyses using the BASCOE transport model**

*Chabrillat et al., ACPD, 2018*

We thank the reviewer for his/her insightful comments. It appears that the version of the manuscript which was reviewed by this referee is the version first submitted to ACPD (on 4 April 2018) rather than the version finally published in ACPD (on 7 May 2018). Fortunately all comments apply equally to both versions. In our replies below the bold type is used to highlight text in the revised manuscript.

**Replies to general comments**

- *The principal conclusion is that the simulations of AoA obtained when BASCOE is constrained by different reanalyzed datasets differ substantially from one another. This is not at all unexpected given the differences among the reanalysis models.*

The reanalysis systems are based on different models but they assimilate very similar satellite datasets. Many users of reanalyses are neither modellers of stratospheric dynamics nor aware of the lack of observational information to constrain the BDC in the reanalyses. From feedback obtained at the 5th International Conference on Reanalysis (ICR5, Rome, November 2017), such users do not expect to see a spread between the reanalyses which is as large as between unconstrained GCCMs (Fig. 4). On Fig. 8 they easily understand that the uncertainties in the observational timeseries are large (due to sparse and irregular sampling) but they do not expect to see that the spread between the reanalyses is as large as these observational uncertainties.

A third highlight of this paper is the intercomparison of AoA trends between the reanalyses. Several reanalysis intercomparisons of diagnostics related to stratospheric dynamics have already been published and showed significant differences with respect to their trends (e.g. Abalos et al., 2015; Miyazaki et al., 2016). Yet for the AoA diagnostic, most recent studies rely on ERA-I with much interest in the latitudinal structure of its trends. We found that over the post-2002 period ERA-I is the *only* reanalysis to deliver opposite trends of AoA in the two hemispheres (Fig. 12, middle column). This is also an unexpected result.

- *The paper is well organized and clearly written, with some exceptions, the main one being that the procedure for computing AoA is not well explained. In particular, it is not clear whether AoA is calculated with respect to a reference level at the tropical tropopause or in the troposphere, and this introduces some ambiguity in the interpretation of the results.*

We agree with the referee that the handling of the reference level was problematic in the submitted manuscript. All our calculations used the surface both as the source region and to compute the time lag defining the mean Age of Air. But in order to better highlight the different transit times from the equatorial tropopause, Fig.1 and 3 were corrected a posteriori by subtraction of the time-averaged AoA at 100hPa, 10°S-10°N. All other figures used the surface as reference, hence including the transit time from the surface to the tropical tropopause. This distinction was not clearly made and led to inconsistent figures, as shown by several specific comments made by both reviewers.

Hence we decided to re-run our calculations and re-plot all figures (except for figure 8, see below) using as source region the tropical tropopause region (still defined as the 100 hPa isobar between latitudes 10°S and 10°N), and computing the AoA at each grid point as the time elapsed since the mixing ratio of the ideal tracer reached the same value in that source region. The last paragraph of section 2.1 has been re-written to fully explain the updated procedure for computing AoA.

The figures did not change much from the discussion paper, indicating that this is a methodological issue which does not have a large impact on our findings. Besides figures 3 and 9 which are discussed below for specific comments, there is one other case where the figure changed sufficiently to warrant a minor update in the text: on figure 12 the positive AoA trends for ERA-I in 2002-2015 (top row, middle column) have become significant at all latitudes (in the discussion paper they were significant only in the polar latitudes). On figure 12 the signs and patterns of AoA trends did not change for any other reanalysis or period but the range of these trends increased by up to 50% (see min/max values above the plots); this led us to extend the scale of the color bar, from [-0.6,0.6] to [-0.9,0.9].

For figure 8 (and figure 8 only) we have kept the original calculations where the tracer was set to increase linearly throughout the surface, because this figure includes a comparison with observational values of AoA which used the surface as reference. We have moved to the discussion of figure 8 the description of this surface boundary condition and its propagation through the troposphere, because it is now irrelevant for all other figures. We have also added in this figure a plot showing tropical AoA computed both from the surface and from the tropical tropopause to show that the difference does not vary significantly with the simulated year (see next comment).

**Replies to specific comments**

- *(4, 20) "There is no other representation of convection": It is not clear that, in the Tropics, where deep convection can reach the 14-15 km level, this artificial diffusion can simulate vertical transport realistically. But perhaps this does not matter for assessments of AoA in the stratosphere if the base point for AoA calculations is taken to be at or near the tropical tropopause? Please comment (especially since it is not clear how the reference level for computing AoA is chosen). See also comment at (7, 24).*

As explained above, nearly all figures now use (100hPa, 10°S-10°N) as reference region hence the absence of deep convection in the CTM is not an issue for the updated figures. Since for figure 8 we keep using the surface as reference, we added tropical timeseries in the tropics which show both the surface-based evaluation (solid lines) and the tropopause-based evaluation (dashed lines):

[Figure]

**Middle panel of revised Fig. 8**. Mean AoA in the mid-stratosphere (5-30hPa) for the tropical latitudes (30°S–30°N). Solid lines show AoA using the surface as reference, dashed lines show AoA using the tropical tropopause as reference (i.e. as in all other figures of the revised manuscript).

This comparison between the two evaluations also allows a discussion on the impact of the omission of deep conviction in the model. The discussion of Fig. 8 in the revised manuscript includes the following paragraph:

> **The differences between the two calculations represent the transit times from the surface to the tropical tropopause, are nearly independent of the simulated year and range between 3 months (with ERA-I or JRA-55) and 6 months (with MERRA). These values are close to the longest transit times reported in a recent intercomparison of global models (Krol et al., 2018), indicating a rather slow transport from the surface to the tropical tropopause which we attribute to the omission of deep convective transport in our model. While the surface-based model AoA (solid lines in Fig. 8) may be slightly overestimated, these biases have no significant inter-annual variations and do not hinder the intercomparison between reanalyses.**

- *(6, 5) "at the wavelength number 47": "at wavenumber 47" might be better.*

Text corrected.

- *(6, 8) "Figure 1 compares the results": I do not believe you have stated how AoA is calculated. [...]*

Section 2.1 now describes explicitly the revised procedure to calculate AoA from the tropical tropopause region:

> **The age of air is defined as the spectrum of transit times from a source region to a given location, with the tropical tropopause usually defining the source region for studies of the stratosphere. In the case of ideal tracers which increase linearly in the source region and have no photochemical production or losses, the mean of this spectrum (denoted here AoA) is simply the time elapsed since the mixing ratio of this ideal tracer reached the same value in the source region (see e.g. Waugh and Hall, 2002). We follow here this classical approach, using for most simulations the 100 hPa isobar between latitudes 10°S and 10°N as source region.**

Section 3.2 describes explicitly the original procedure which has been kept only for figure 8:

> **For consistency the modeled AoA in this figure are evaluated as the time elapsed since the mixing ratio of an ideal tracer reached the same value at the surface, using as boundary condition a global constant increasing linearly with time at the surface.**

- *(7, 18) Figure 2: This figure, as well as Figure 3, would benefit from a color bar to indicate the values of the AoA isolines not explicitly labeled.*

A color bar has been added to figures 2 and 3.

- *(7, 24) Figure 3: I am confused by this comparison [...] So, where is the reference point in these simulations? If it is at the surface, then AoA will reflect the effects of transport not just in the stratosphere, but also in the troposphere, including the artificial diffusive transport between the surface and the middle troposphere. Unless I am misunderstanding what you have done here, it seems to me that, if AoA is intended to highlight transport in the stratosphere (e.g., Waugh and Hall, 2002, Sec. 3.1) then the choice of a base point in the troposphere confuses the issue, especially given the use of artificial diffusive transport in the lower troposphere.*

The reviewer was rightly confused and his interpretation is correct. We have followed this advice, choosing the tropical tropopause as reference point in the revised manuscript (see above). The relative differences between ERA-I and the four other reanalyses vanish at the reference point and the difference is not plotted at grid points where ERA-I AoA is smaller than 5 days.

- *(8, 5) "The intercomparison at 50 hPa": You should state explicitly in the text that in these comparisons AoA is "normalized" to be zero at the tropical tropopause (this is only stated in the caption of Figure 4). Otherwise, the reader will wonder, as I did, why the AoA shown in Figures 2-3 are different from the AoA in Figure 4. By the way, a problem with the "normalization" of AoA to zero at 100 hPa is that it gives the impression that AoA above that level is determined only by the stratospheric circulation, when in fact the AoA also contains the effect of transport in the troposphere.*

thanks to the direct calculation of AoA using the tropical tropopause as reference point, no "normalization" is performed any more for the figures 1 and 4 of the revised manuscript.

- *(8, 12) "overall, the spread . . . is larger than the 1-sigma. . .": One wonders how this result would change if AoA were computed with respect to a reference point at 100 hPa.*

In the revised manuscript the AoA are computed with respect to a reference point at 100 hPa. The differences in Figure 4 between the submitted and revised manuscripts are nearly indistinguishable. Hence the spread between the five simulations at 50 hPa is still larger than the 1-σ observational uncertainties in the tropics, and still nearly as large in the extratropics. We have not modified this sentence in the revised manuscript.

- *(8, 26) "The spread between the four reanalyses reaches a maximum of 0.2 years at 30 hPa": Are you referring here to the gradient comparison, Figure 4d? How is this "gradient" calculated? The figure legend refers to "MLNH-Tropics" and shows values in months, not per unit distance, so this is really a difference between the Tropics andmidlatitudes o f the NH. How are Tropics and NH midlatitudes defined?*

The words "(mean age) gradient profiles" or "latitudinal gradients (of mean age)" were meant with the same meaning as Neu et al. (2010) and Chipperfield et al. (2014) i.e. as the difference between AoA in NH midlatitudes and AoA in the Tropics. The vertical profiles on figure 4d simply show the differences between the corresponding profiles on figures 4c and 4b which are mean values for latitude bands 35°N-45°N and 10°S-10°N respectively (as stated in the figure of caption 4).

In the revised manuscript we have added the definition of the latitude bands in the discussion of figures 4b and 4c and we have added the following sentences in the discussion of figure 4d:

> **These "latitudinal gradients of AoA" were used in several CCM intercomparisons (Neu et al., 2010; Chipperfield et al., 2014). Figure 4d shows this diagnostic for the five reanalyses, i.e. the differences between the AoA profiles on Fig. 4c and Fig. 4b.**

We have replaced the words "latitudinal gradients" by "AoA differences" in the remainder of this discussion and in the caption of the figure.

- *(8, 30) "MERRA-2 yields an outlying vertical profile of AoA at northern midlatitudes": True with respect to the other reanalyses except for MERRA (Fig. 4c), and in fact, MERRA and MERRA-2 midlatitude profiles of AoA agree best with the observations. You keep referring to MERRA-2 as an "outlier", which carries negative connotations,but in fact being an outlier in this comparison is a good thing if one considers the data to be the "truth"..*

Thanks for pointing this out. We have corrected the discussion according to your comment:
**MERRA and MERRA-2 yield larger AoA at northern midlatitudes than the three other reanalyses. In the case of MERRA-2 this results in a profile of AoA differences which are significantly larger than the profiles obtained with the four other reanalyses but agrees much better with the profile derived from the observations. Hence MERRA-2 apparently underestimates the tropical upwelling in the lowermost stratosphere (100-60~hPa), agrees better with the observations at 50~hPa than any other reanalysis, and joins the results of the four other reanalyses at higher levels.**

- *(9, 14) "MERRA-2 starts with much older values": This behavior does appear to be anomalous. Any idea what might be causing it?*

This issue is discussed in detail in sectin 5 (see paragraph starting with "MERRA-2"). We have inserted the following sentence in section 3.2:
**The possible causes for this apparently anomalous behavior of MERRA-2 are discussed in section 5.**

- *(9, 18) "The Pinatubo eruption does not appear to have any impact of the simulated AoA at 50 hPa": Insofar as one might expect that the largest impact of Pinatubo would be in the Tropics, it might be worthwhile to examine the AoA time series averaged over, say, 30N-30S.*

This was done and no impact was found for the tropical latitude band, as shown by the corresponding plot:

[Figure]

Note that any impact in the 30°S-30°N latitude band would have been seen in figure 7 which shows the 72°S-72°N band. The revised manuscript mentions the absence of volcanic impact in the tropical latitude band as well.

- *(10, 8) "observational trend is not significant": One would not expect any trend calculated from the smoothed, model time series shown on the right pane of Figure 8 to be significant either. By the way, you keep referring to "trends" in connection with the model results, but you have not calculated any trends...*

Not yet. As outlined in the introduction, modelled trends are evaluated extensively in section 4. The sentence now refers to this later finding:

> **ERA-I delivers a weakly positive trend over the period 1989-2015 and we will assess in section 4.3 that this trend in the model results is significant.**

And this finding has been added in the discussion of figure 12 (section 4.3):

> **The same plot** (*i.e. Fig. 12, top right)* **also shows that the positive trend which had been inferred visually for the northern mid-latitudes of the middle stratosphere (Fig. 8, left) is significant.**

- *...Note also that Garcia et al. (2011) have argued that, even using model output for an ideal AoA tracer, trends over periods as long as 30 years are often not significant when the ideal tracer is sampled like the available observations of stratospheric tracers....*

Great caution should indeed be exercised in comparisons of trends between model output and observational datasets which are sparsely and irregularly sampled. This is what we meant in the original manuscript with "..., but Engel et al. (2009, 2017) showed that the sign of this observational trend is not significant". Referring to Garcia et al. (2011) allows us to reinforce and clarify this call to caution. The revised manuscript states:

> **While the overall trend simulated with ERA-I is apparently in agreement with the balloon observations, this comparison should be considered with great caution because the sign of the AoA trend is not significant in the observations (Engel et al., 2009, 2017) and modelled trends over periods as long as 30 years are often not significant when the ideal tracer is sampled like the available observations of stratospheric tracers (Garcia et al., 2011).**

- *... Furthermore, trends derived from observation are also confounded by the fact that no real atmospheric tracer has a constant, linear growth rate.*

The non-linearity of the growth rate of $CO_2$ and $SF_6$ has of course been taken into account by Engel et al. (2009, 2017) for their derivation of AoA and also for their error analysis. The procedure is described in the supplementary information of Engel et al. (2009) and in the first paragraph of section 4 in Engel et al. (2017).

- *(11,16) "in the polar regions and midlatitudes": Why limit this to extratropical behavior only? It would be interesting to show the seasonal amplitude in the Tropics as well, say a ±30° average.*

Thanks for the suggestion. The seasonal amplitudes with ERA-I and MERRA-2 have a different vertical structure in the Tropics. We have added this plot to figure 9 in the revised manuscript.

- *(11, 24) "MERRA and MERRA-2 . . . different amplitudes depending on the period used or the analysis": Does this have to do with the development (1989-2001) and stabilization/decline (2002-2015) of the Antarctic ozone hole? To explore this issue, one would have to examine the actual seasonal climatology at high SH latitudes, not just the annual amplitude.*

The left plot below is extracted from fig. 9 in the ACPD manuscript while the right plot shows results with the new AoA calculation (i.e. using the tropical tropopause as reference):

[Figure]

This shows that the new AoA calculation delivers amplitudes of AoA seasonal variations which are much closer for periods 1989-2001 and 2002-2015. This is the case also for the other latitude bands, so for the revised manuscript we have removed from figure 9 the results for the early period 1989-2001 and we have dropped the corresponding part of the discussion.

- *(12, 9) "could not be expected from inspection of the native dynamic variables": I do not understand what you are trying to say here. Please elaborate. And note also that the discrepancies you mention ("up to 50% dependencies on the considered time period") are not even illustrated, so it is very difficult to even guess what the intent of your statement is.*

The dependencies of seasonal amplitudes on the considered time period reached 50% in fig. 9a and 9c of the ACPD manuscript but, as explained in the previous comment, they have disappeared from the revised manuscript thanks to the corrected AoA calculation. We removed the whole paragraph (i.e. last paragraph of section 4.2) from the revised manuscript because it is redundant with the second paragraph of section 5.

- *(12, 15) "unexpectedly increasing": Given the very short period covered by the $SF_6$ observations, it is not clear that one should "expect" any particular sign for the trends. Determination of AoA trends from observations of stratospheric tracers is fraught with many uncertainties; even in models where an ideal, linearly increasing artificial tracer is used, one has to rely on zonal-mean results over long periods to obtain trends that are clearly statistically significant. Arguably, examination of AoA trends determined from observations of stratospheric tracers is not the best tool for documenting changes in the BD circulation. See Garcia et al. (2011).*

We acknowledge that it is important to provide proper context about this topic. Section 4.3 now starts with a new paragraph explaining some caveats of interpreting AoA trends as changes in BDC. The next paragraph has been expanded and corrected to justify the interest of the section:

**It is delicate to infer changes in the BDC on the basis of AoA trends over periods shorter than several decades. Even in models where an ideal, linearly increasing artificial tracer is used, one has to rely on zonal-mean results over long periods to obtain trends that are clearly statistically significant (Garcia et al., 2011). The statement that is often made that climate models simulate a decreasing AoA throughout the stratosphere only applies over long time periods and is not necessarily the case for the past 25 years, when most tracer measurements were taken (Garfinkel et al., 2017). For example, the analysis of a 1700 year simulation showed that it takes around 30 years for a modeled BDC trend to emerge from the noise of natural climate variability (assuming a 2%/decade trend in the BDC; Hardiman et al., 2017).**

**While linear trends of AoA over shorter periods may represent transient changes due to climate variability, such changes over timescales which are intermediate between the QBO and the multidecadal scales are still relevant to the study of stratospheric dynamics.** Current research on AoA trends has largely focused on a dipole-like latitudinal structure for the period 2002-2012, which was first derived from satellite observation of $SF_6$ by the MIPAS instrument (Stiller et al., 2012). This structure of trends shows AoA decreasing in the Southern Hemisphere but  increasing in the Northern Hemisphere **which was used to explain a recent increase of stratospheric HCl in the Northern Hemisphere (Mahieu et al., 2014) and interpreted as the** consequence of a southward shift of the subtropical transport barriers (Stiller et al., 2017) .

- *(13, 1) "unexpected growth": Again, there are no clear expectations about trends for short periods.*

We have removed the word "unexpected" from this sentence.

- *(13, 23) "the reversal is found for all five reanalyses": This is true, but the reversals are in the opposite sense in ERAi and CFS vs. JRA-55, MERRA and MERRA2, so it is hard to know what to make of this.*

Yes but we still want to highlight this striking feature in case some reader does find what to make of this. We have changed this sentence to:

This reversal is found for all five reanalyses and in all regions of the stratosphere **but it is difficult to interpret because it goes in opposite directions in ERA-I and CFSR versus JRA-55, MERRA and MERRA-2.**

- *(13, 25) "unexpectedly large disagreements": I am not sure why you think the disagreements are "unexpected". While presumably all reanalyses use more or less the same observational data, the manner in which the data are assimilated and the physical parameterizations included in the different models (in particular, those for convection and mesoscale gravity waves) are different. Note that at (14, 8) you suggest that "the disagreements found here may lie in the differences between the underlying models"; I agree that this is the most plausible working hypothesis.*

See above the reply to the first general comment. The word "unexpectedly" had already been removed before publication in ACPD.

**Grammar, typos, etc.**

All errors have been corrected.

**Added references**

[revised manuscript text omitted]
 an ideal tracer which increases linearly  in the source region and has no photochemical productions or losses, one can obtain the mean of this spectrum (denoted here AoA) at any time and location from the corresponding mixing ratio of the tracer: in such a case the AoA is simply the time elapsed since the ideal tracer had the same mixing ratio in the source region (Waugh and Hall, 2002). We follow here this classical approach, using for most simulations the 100 hPa isobar between latitudes 10°S and 10°N as source region. In one case we have used the surface as source region in order to enable a comparison with a long time series of balloon observations (see section 3.2). The output AoA datasets are interpolated from model levels to constant pressure levels using the instantaneous and two-dimensional input surface pressures, i.e. prior to any averaging in the longitudinal or time dimension.

~~In order to allow quick propagation of this boundary condition to the free troposphere, eddy vertical diffusion is modeled in the lower half of the troposphere with a vertical diffusion coefficient $K_{zz}$ decreasing from an arbitrary value of 10 m² s⁻¹ at the surface to zero at the pressure level halfway between the surface and the tropopause.There is no other representation of convection in the model nor any explicit mechanism for horizontal diffusion.~~

**2.2 Description of the input reanalyses**

We compute and compare the AoA in five recent reanalyses which are described in detail by Fujiwara et al. (2017): ERA-Interim  (European Centre for Medium-Range Weather Forecasts Interim Reanalysis; Dee et al., 2011), JRA-55  (Japanese 55-year Reanalysis; Kobayashi et al., 2015), MERRA (Modern Era Retrospective-Analysis for Research; Rienecker et al., 2011), MERRA-2 (Gelaro et al., 2017) and NCEP-CFSR  (National Centers for Environmental Prediction–Climate Forecast System Reanalysis; Saha et al., 2010). These data-sets were used over the period January 1980 to December 2015, except for NCEP-CFSR which originally ended in December 2010 and is extended here with the CFSv2 data-set (Climate Forecast System version 2 Saha et al., 2014) from January 2011 to December 2014. Hereafter we use "ERA-I" to refer to ERA-Interim and "CFSR" to refer to the combined NCEP-CFSR reanalyses.

Each reanalysis is available on two vertical grids: the native grid of the underlying atmospheric model (product on "model levels") and an output grid of constant pressures (product interpolated to "pressure levels"). Our simulations are run on the native model levels in order to account for the different vertical resolution of each reanalysis system and also to avoid any interference from the interpolation methods used to deliver the products on constant pressure levels. All reanalysis systems use the hybrid sigma-pressure vertical coordinate with levels extending from the surface up to ~0.266 hPa (~57 km height) in CFSR, 0.1 hPa (~64 km) in ERA-I and JRA-55, or 0.01 hPa (~78 km) in MERRA and MERRA-2. The reader is referred to Fujiwara et al. (2017) for a comparison of the vertical resolutions of the reanalysis systems.

The forecast models use two different frameworks to discretize their primitive variables on the horizontal plane: MERRA and MERRA-2 solve for mass fluxes on a regular latitude-longitude grid (Lin, 2004) while ERA-I, JRA-55 and CFSR use spectral dynamical cores, i.e., they solve for vorticity and divergence expressed on a spherical harmonics basis (e.g., Krishnamurti et al., 2006). Users of the reanalyses often download velocity fields which are derived from the primitive variables and evaluated on

5   varying regular grids: these may be reduced Gaussian grids (ERA-I and JRA-55), regular Gaussian grids (CFSR) or regular latitude-longitude grids (MERRA and MERRA-2). This pre-processing is described in detail in the next subsection.

We use in all cases the analyses valid at 00 h, 06 h, 12 h and 18 h, i.e., datasets with a 6-h time resolution. The assimilation procedure for MERRA and MERRA-2 uses an iterative predictor–corrector approach, generating two separate sets of reanalysis products designated "ANA" for analysis state and "ASM" for assimilated state (Rienecker et al., 2011). The latter products use

10   a 6h "corrector" forecast centered on the analysis time and an incremental analysis update to apply the previously calculated assimilation increment gradually rather than abruptly at the analysis time (Bloom et al., 1996). Thanks to this procedure, the ASM products have smaller wind imbalances than the ANA products (Fujiwara et al., 2017) hence they are preferable for tracer transport simulations. We used the ASM products in MERRA-2 but could not do so with MERRA where the ASM products are only available on constant pressure levels. Since we aim to evaluate each reanalysis on its native vertical grid, we had to

15   fall back on the ANA product in the case of MERRA.

**2.3 Pre-processing of the reanalyses**

 The BASCOE Transport Model (hereafter BASCOE TM) is used as a tool to perform a fair comparison of advective transport in each reanalysis data-set, using their native vertical grids but a common, low-resolution latitude-longitude grid. It requires on input the surface pressure and horizontal velocity on a so-called Arakawa C-grid, i.e.,

20   the zonal wind $u$ must be staggered in longitude and the meridional wind $v$ must be staggered in latitude. As indicated by its name, the FFSL algorithm evaluates internally the corresponding mass fluxes and derives the vertical winds ($w$) from mass conservation. Hence the reanalysis datasets must be carefully pre-processed from spectral or high-resolution gridded fields to the low-resolution C-grid. We have paid special attention to this pre-processing of the reanalyses to make sure that the different types of wind fields are expressed in a consistent manner for our transport algorithm.

25   Due to its assimilation procedure, the early ERA-40 reanalysis contained large dynamical imbalances which deteriorated the Brewer-Dobson circulation through excessive upward motion in the tropics and excessive transport from the tropics to the mid-latitudes (Meijer et al., 2004; Monge-Sanz et al., 2007). Pawson et al. (2007) described a similar issue with MERRA and proposed to use time-averaged input wind fields in order to remove these imbalances, but this approach is available only for MERRA and MERRA-2.  To filter out such dynamical imbalances, BASCOE uses a pre-processor

30   which was originally developed only for the analyses computed by the European Centre for Medium-Range Weather Forecast (ECMWF) including ERA-I (Segers et al., 2002; Bregman et al., 2003) . 
[revised manuscript text omitted]

Figure 12 compares the latitude-pressure distributions of AoA trends across all five reanalyses and for the early (1989-2001), recent (2002-2015) and overall periods (1989-2015). It is important to note that the trends over the early and overall periods

15  should be considered with caution since there was little data to constrain the stratospheric winds until 1998 (see the discussion in the next section). The AoA trends derived from ERA-I wind fields during the early period (Fig. 12, upper left)  grow in both hemispheres  except for the northern lowermost stratosphere. During the recent period, the dipole structure derived from ERA-I (Fig. 12, upper middle) is similar  than over the slightly shorter period 2002-2012 (Fig. 11) . The increases in the

20   Northern Hemisphere become weaker but remain significant at all latitudes, although at fewer grid points. The maximum trend is located at 24°N and 25 hPa where it slightly exceeds $0.3 \pm 0.2$ years per decade. 
[revised manuscript text omitted]

30    reaching about 0.5 years per decade. The increase in the Northern Hemisphere is significant (at the 95% confidence level) and it is not obtained in multidecadal climate model simulations. Yet the trends derived from ERA-I are shown to closely depend on the considered period. When it is extended to 2002-2015, the positive trends in the Northern Hemisphere become weaker (about 0.3 years per decade) and they are significant at fewer grid points. A further extension to 1989-2015

35   shows  that the negative trends in

the southern middle stratosphere become insignificant. For all five reanalyses the trends over the early period (1989-2001) have opposite signs than over the recent period (2002-2015). Looking only at the recent period which is better constrained by observations, the main outcome is again large disagreements between the reanalyses: JRA-55, MERRA and MERRA-2 provide increasing AoA in the middle stratosphere while CFSR provides a decreasing but mostly insignificant trend. To summarize, the signs of the trends depend strongly on the input reanalysis and on the considered period with values above 10 hPa varying between approximately -0.4 and 0.4 years per decade. Independently of the considered period, no reanalysis other than ERA-I finds any dipole structure in the latitude-height distribution of AoA trends.

No Since the wind fields are weakly constrained, the causes for the disagreements found here may lie in the differences between the underlying models. While no obvious cause could be found for these disagreements. The , we suggest that the parametrization of non-orographic gravity wave drag in the underlying dynamical model deserves further investigation, especially in the case of MERRA-2 which has difficulties to represent correctly the QBO before 1995. No global impact of the Pinatubo eruption can be found in our simulations of AoA, contrarily contrary to a recent study which used ERA-I and JRA-55 to drive a diabatic transport model. This highlights the need to repeat the present intercomparison with diabatic transport models because they would reflect directly the significant differences between the heating rates in the reanalyses (Wright and Fueglistaler, 2013). Future work will also focus on quantitative comparisons with AoA derived from MIPAS observations of $SF_6$; comparisons with the long-term records of other long-lived tracers to provide further insight at multidecadal scales; and disentangling the contributions to AoA of residual circulation, mixing on resolved scales and mixing on unresolved scales.

The main conclusion of this study is the significant diversity in the distribution of mean AoA which we obtain with our transport model, depending on the input reanalysis. This casts doubt on our ability to model accurately the time necessary for variations of greenhouse or ozone-depleting species to propagate from the troposphere to the stratosphere. We have also found large disagreements between the five reanalyses with respect to the long-term trends of age of air. This suggests that with our type of offline transport model, the wind fields in modern reanalyses are not sufficiently constrained by observations to evaluate the actual changes of stratospheric circulation. Yet this conclusion should not be hastily extended to other types of transport models which also use the reanalyses of temperature and heating rates.

*Code and data availability.* The monthly zonal averages of AoA, as delivered by the BASCOE TM experiments driven by the five input reanalyses, are distributed as an online supplement to this article. The source code of the BASCOE TM, including its tools to pre-process the reanalyses, is available by email request to the corresponding author. The ERA-Interim reanalysis (Dee et al., 2011) is provided by the ECMWF, see http://www.ecmwf.int/en/forecasts/datasets. MERRA data (Rienecker et al., 2011) and MERRA-2 data (Gelaro et al., 2017) are provided by the Global Modeling and Assimilation Office at NASA Goddard Space Flight Center through the NASA GES DISC online archive; see https://disc.gsfc.nasa.gov/information/glossary?keywords=merra. The CFSR (Saha et al., 2010) and CFSv2 (Saha et al., 2014) reanalyses data were obtained from NOAA NCEP; see http://cfs.ncep.noaa.gov/. The JRA-55 reanalysis (Kobayashi et al., 2015) was obtained from the NCAR Research Data Archive; see https://rda.ucar.edu/datasets/ds628.0/.

*Acknowledgements.* We thank the reanalysis centers (ECMWF, NASA GSFC, NOAA NCEP and JMA) for providing their support and data products. We thank Dr. Gabriele Stiller, Dr. Paul Konopka and Dr. Bernard Legras for fruitful discussions during the preliminary steps leading to this study, and Dr. Masatomo Fujiwara for his coordination of the S-RIP. We would also like to thank the editor and two anonymous reviewers for their valuable comments. Yves Christophe's contribution was partly supported by the European Commission project MACC-II under the EU Seventh Research Framework Programme (contract number 283576). Daniele Minganti's contribution was financially supported by the Fonds de la Rechecherce Fondamentale Collective through research project ACCROSS (convention PDRT.0040.16). Emmanuel Mahieu is Research Associate with the F.R.S.-FNRS.

[revised manuscript text omitted]